# ONE-FOR-ALL: TOWARDS HUMAN-CENTRIC MULTI-SUBJECT CUSTOMIZATION FROM SINGLE-SUBJECT EXAMPLES

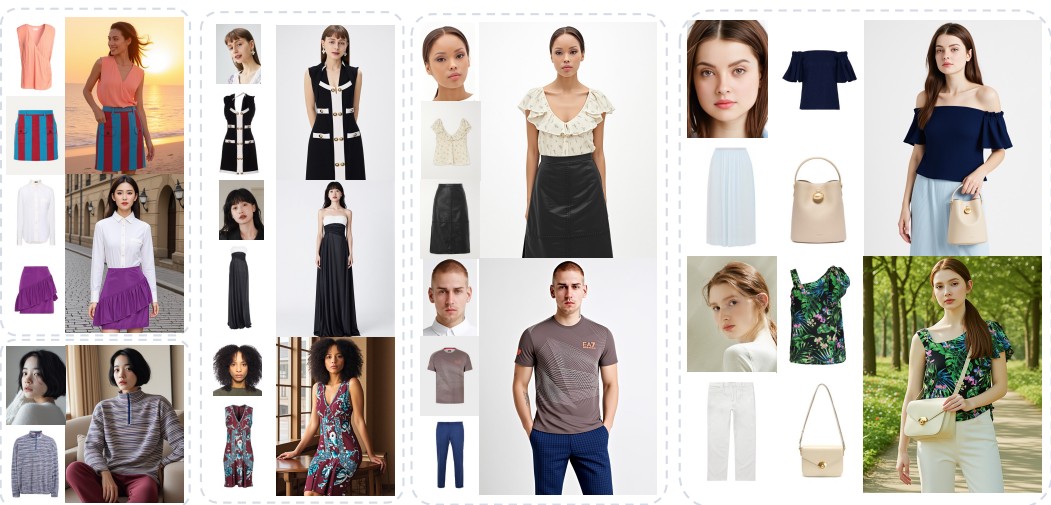

Figure 1: Visualization results of our One-for-All for human-centric multi-subject customization. Trained solely on single-subject data pairs, our approach supports flexible combinations of reference subjects. It demonstrates high-fidelity generation effects for faces of various races, garments with diverse styles and complex textures, and bags with different designs.

## ABSTRACT

Human-centric multi-subject customization remains a key challenge in the field of subject-driven image synthesis. A primary obstacle lies in the curation of paired multi-subject data, which is labor-intensive and often introduces subject inconsistencies that hinder effective model learning. In this paper, we introduce One-for-All, a framework that pioneers a new paradigm by learning multi-subject consistency from only real-world, single-subject examples, breaking the dependency on curated multi-subject data. Building upon this, we unlock the full potential of this paradigm shift by introducing two key designs that ensure robust multi-subject consistency. Firstly, a Center-Aligned Cross-Modal Position Association module is proposed to guide the interaction between visual references and their textual descriptions. This interaction facilitates intra-subject semantic grounding among cross-modal conditions and improves their synergistic contributions for subject consistency. Secondly, to alleviate the attention dilution caused by the increased tokens of multiple subjects, a Dynamic Attention Modulation mechanism is introduced. This design maintains multi-subject consistency by dynamically predicting and applying token-wise attention weights, ensuring focus remains on critical features. Comprehensive experiments demonstrate that our method, even trained exclusively on single-subject data, exhibits robust generalization across varying numbers of reference subjects, and surpasses all baseline methods trained on curated multi-subject data pairs.

# 1 INTRODUCTION

Recent advances in generative models (Rombach et al., 2022; Esser et al., 2024; Labs, 2024; Labs et al., 2025) have pivoted the research focus towards fine-grained controllable generation, especially for personalized contents. Human-centric customization, in particular, has garnered interest for its potential to unlock novel applications, e.g., virtual try-on, character design, etc. Despite progress, a critical challenge persists. Existing methods (Ye et al., 2023; Wang et al., 2024a; Huang et al., 2024; Wang et al., 2025) excel at customizing a single subject, such as a face or a garment. However, the complexity escalates for multi-subject human generation, which involves creating a coherent image from distinct user-provided subjects like a specific face, particular garments, and a unique accessory.

Notably, although existing single-subject driven approaches (Li et al., 2024b; Chen et al., 2024; Tan et al., 2024) have proven effective for condition learning from single-subject datasets, their extension to multi-subject settings remains challenging, primarily due to the difficulty of acquiring high-quality multi-subject data. Some methods (Wang et al., 2024b; Fan et al., 2025; Xiao et al., 2025; Chen et al., 2025) extract multi-subject data pairs from images or videos. However, they rely on cumbersome data processing steps, e.g., identification, segmentation, and cropping. This process is not only time-consuming but also results in limited data diversity, which easily leads to overfitting. Another line of work (Wu et al., 2025b; Mou et al., 2025) curates multi-subject data via conditioned generation. While this workflow yields visually plausible results, it suffers from limited consistency across generated pairs. As the number of reference subjects increases, maintaining this consistency becomes nearly impossible, which in turn limits the model's ability to generate coherent contents. As shown in Figure 2, UNO (Wu et al., 2025b) struggles to model fine-grained details on face and clothing when learning from curated data pairs via generation, due to inconsistencies in the synthetic image pairs. Therefore, an intuitive yet powerful idea arises: *to leverage the easily collected high-quality, single-subject data for the challenging task of multi-subject customization.*

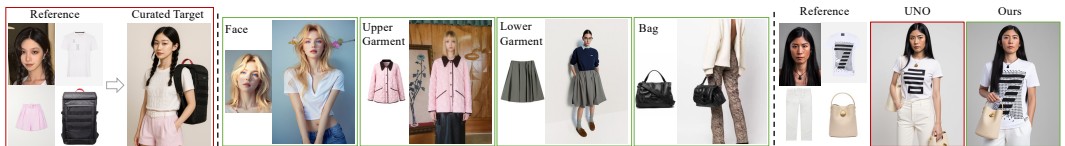

Figure 2: Comparison of data requirements and generation fidelity. Red box: Curated multi-subject training pairs via generation and synthetic results from UNO (Wu et al., 2025b). Green box: High-fidelity, single-subject training pairs and results from our method. Our method excels at simultaneously maintaining high fidelity for faces, garment textures, and bag details.

One direct implementation involves randomly selecting a single reference type to condition the model during training, and then leveraging multiple references at inference time. However, this discrepancy between the single-subject training paradigm and the multi-subject inference introduces another challenge. Concretely, the standard full-attention mechanism shows inadequate in this setting, leading to two critical issues: On one hand, this leads to semantic grounding ambiguity (*Text-vs-Image*). The model, accustomed to a one-to-one mapping in training, struggles to perform targeted text-to-visual concept association when multiple subjects are present, risking failures like attribute leakage or consistency degradation. On the other hand, it creates representational contention (*Image-vs-Image*). The concurrent introduction of multiple references leads to competition for the limited attention capacity, causing attention dilution. As a result, the model's focus is divided, simultaneously degrading the fidelity and consistency of each subject. A robust model must therefore possess a mechanism for conflict resolution that ensures precise semantic grounding while preserving the representational integrity of each subject.

To tackle the above challenges, we propose a novel human-centric multi-subject customization framework, named One-for-All. Our key insight lies in learning robust multi-subject consistency by leveraging only single-subject real-world data, circumventing the need for curated multi-subject datasets. Specifically, we assign distinct condition IDs to various human-centric subjects (face, upper garment, lower garment, etc.) and enforce a learning process during training, guiding the model to associate each subject category with its specific position. Building on this foundation, we introduce two novel mechanisms to adapt to paradigm shift and ensure multi-subject consistency: 1) A

Center-aligned Cross-modal Position Association (CCPA) module that establishes an explicit position correspondence between textual descriptors and their respective visual concepts. This ensures that cross-modal interaction is precisely targeted, facilitating intra-subject semantic grounding and synergistic contributions of multi-modal conditions. 2) A Dynamic Attention Modulation (DAM) mechanism that predicts token-wise attention weights from the noisy latent query. This allows the model to adaptively adjust the influence of each reference subject, thereby alleviating feature conflicts and ensuring robust multi-subject consistency.

For experimental evaluation, we introduce HumanBench, a benchmark of diverse human face and clothing pairs. Comprehensive comparative analyses are conducted against state-of-the-art methods. Both qualitative and quantitative results demonstrate the effectiveness of our proposed approach. **The main contributions can be summarized as follows: (1)** We present a novel framework that learns robust multi-subject consistency by leveraging only real-world, single-subject data. The core of our method is a position-aware learning process that introduces distinct condition IDs for human-centric subjects, which forces the model to associate each subject category with its specific spatial location. **(2)** Two novel mechanisms are proposed to adapt to paradigm shift and ensure multi-subject consistency. The CCPA module first establishes explicit position correspondences between text and visual concepts. Then the DAM module predicts dynamic, token-wise attention weights to adaptively control each subject's influence. **(3)** Extensive comparisons with state-of-the-art methods on our HumanBench demonstrate the superiority of the proposed method. Our approach excels in synthesizing highly consistent, human-centric contents from multiple references and exhibits robust generalization across varying numbers of reference subjects.

## 2 RELATED WORK

### 2.1 MULTI-SUBJECT CUSTOMIZATION

Customized image generation aims to synthesize novel images of reference subjects. Early methods (Gal et al., 2022; Ruiz et al., 2023; Wei et al., 2023) target single-subject customization, while later works (Kumari et al., 2023; Hu et al., 2025; Wu et al., 2025b; Mou et al., 2025) extend to multi-subject settings. UNO (Wu et al., 2025b) uses generative models to curate training pairs for different subject combinations but suffers from limited consistency due to the low-quality synthetic data. DreamO (Mou et al., 2025) integrates diverse data sources, yet still produces over-smoothed and inconsistent results. OmniGen2 (Wu et al., 2025a) uses authentic data pairs for better textures but struggles to maintain consistency when handling multiple subjects. MUSAR (Guo et al., 2025) constructs dual-subject pairs from single-subject data. Despite showing a promising direction, scaling to more subjects remains difficult. Overall, two bottlenecks persist: reliance on curated multi-subject data and the challenge of preserving consistency across multiple references.

### 2.2 CONTROLLABLE HUMAN IMAGE GENERATION

For controllable human image synthesis from multiple references, a critical challenge lies in generating high-fidelity textures while modeling complex interactions and proper layering. One representative category of methods focuses exclusively on garment synthesis (Neuberger et al., 2020; Zhang et al., 2024b; Li et al., 2024a), often referred to as multi-garment virtual try-on. While these methods are capable of generating reference-based results for multiple garments across images (Zhu et al., 2024; Velioglu et al., 2025), videos (He et al., 2024), and even 3D scenes (He et al., 2025), their performance is typically constrained by available dataset annotations, leading to fixed training and inference paradigms. Moreover, as these methods are not designed to support face references, their scope of application is limited. More recent approaches supports more diverse reference categories but face persistent limitations. Parts2Whole (Fan et al., 2025) segments faces and garments to form training pairs, restricting composition diversity and causing overfitting. BootComp (Choi et al., 2025) uses a decomposition network for garment pairs. However, the reliance on a separate model poses limited scalability and generalization. Moreover, without explicit differentiation of references, it learns entangled representations, leading to attribute leakage and degraded consistency. Thus, achieving robust multi-subject consistency requires bypassing curated multi-subject data while effectively integrating diverse reference features.

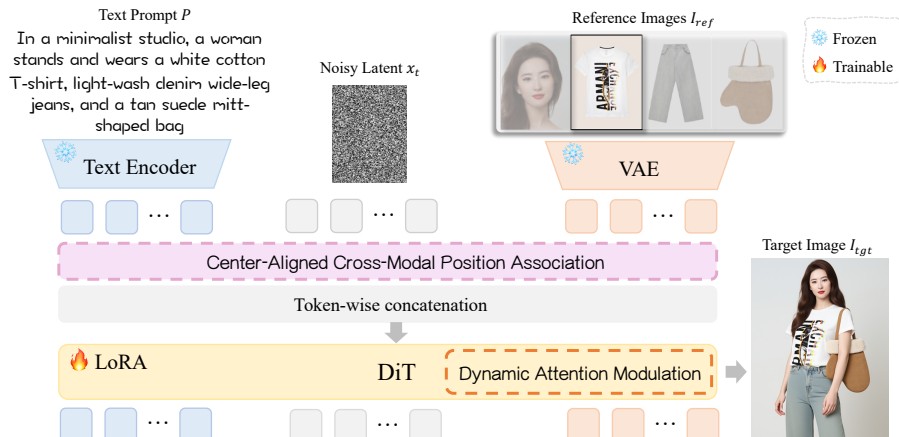

Figure 3: Overall pipeline of One-for-All. Based on the position-aware conditioning strategy, two novel designs to ensure multi-subject consistency are introduced. The center-aligned cross-modal position association module assigns appropriate positional coordinates to the prompt $P$, the noisy latent $x_t$, and the references $I_{ref}$, establishing correlations between cross-modal textual and visual conditions. The dynamic attention modulation module predicts token-wise attention weights, adaptively modulating attention distribution for multiple references.

## 3 METHOD

To circumvent the need of curated multi-subject data pairs and ensure robust multi-subject consistency, we present One-for-All, a framework designed for human-centric multi-subject generation. Our approach's key insight is that multi-subject consistency can be learned from widely available single-subject data by enforcing a novel form of category-specific position conditioning. In this section, we will first review the preliminaries of latent diffusion models (Sec. 3.1). Then, the problem formulation and our core learning strategy will be described (Sec. 3.2). Finally, we will detail the two mechanisms specifically developed to accommodate paradigm shift: the Center-Aligned Cross-Modal Position Association module (Sec. 3.3) to address semantic grounding ambiguity and the Dynamic Attention Modulation module (Sec. 3.4) to manage the visual feature contention.

### 3.1 PRELIMINARIES

The backbone architecture for diffusion models has transitioned from the traditional U-Net (Rombach et al., 2022) to more scalable and powerful Transformer-based designs. Our work builds upon the recent FLUX.1-Kontext (Labs et al., 2025) model, a generative flow matching (Lipman et al., 2022) model that unifies image generation and editing. This model deploys Multi-Modal Diffusion Transformers (MM-DiT) (Esser et al., 2024) architecture. Basically, the initial MM-DiT blocks in Flux.1 (Labs, 2024) treat the noisy image $x_t$ and prompt $P$ with unified Multi-Modal Attention (MM-Attention) layers. In the FLUX.1-Kontext model, a reference image $I_{ref}$ serves as an additional input. Specifically, noisy image tokens and prompt embeddings are first equipped with position embeddings and projected, then concatenated along the token dimension, acquiring query $Q$, key $K$, value $V$. The MM-Attention operation can be expressed as

$$\text{MM-Attention}([P; x_t; I_{ref}]) = \text{softmax}\left(\frac{QK^T}{\sqrt{d}}\right)V, \tag{1}$$

where $[P; x_t; I_{ref}]$ denotes the concatenation of $P$, $x_t$, and $I_{ref}$, $d$ is the dimension of $Q$ and $K$.

### 3.2 PROBLEM FORMULATION

As shown in Figure 3, given text prompt $P$ and a set of $k$ references $I_{ref} = \{I_{ref}^1, I_{ref}^2, ..., I_{ref}^k\}$, our goal is to denoise the Gaussian noise $x_t$ and synthesize a target image $I_{tgt}$. $I_{tgt}$ should preserve the visual fidelity of $I_{ref}$ while maintaining semantic alignment with the prompt $P$. To this end,

the most common solution involves learning consistency from explicitly paired multi-subject data. However, as discussed above, this paradigm faces challenges that data pairs are always difficult to acquire and suffer from limited quality. As a result, overfitting and a low upper bound on achievable consistency are inherent and unavoidable. Therefore, we diverge from this data-centric paradigm and propose a novel solution. Our method learns to compose subjects by assigning fixed, category-specific condition IDs and proposing explicit position encoding for each reference type, as illustrated in Figure 4. Specifically, to differentiate among various subject categories, a unique condition ID is assigned to each (e.g., face: 1, upper garment: 2, lower garment: 3, and bag: 4). Inspired by works (Tan et al., 2024; Mou et al., 2025; Wu et al., 2025b), we map each unique ID to a distinct position coordinate with a diagonal offset. While previous works (Mou et al., 2025; Li et al., 2025) rely on index embeddings to distinguish different subjects, our approach introduces a strong spatial inductive bias, offering a more concrete and distinguishable signal than abstract and optimization-reliant index embeddings. The proposed position-aware conditioning enables the model to extract and integrate distinct reference features based on their pre-defined spatial locations, thereby allowing our training paradigm to learn from randomly sampled single-subject pairs across any category.

### 3.3 CENTER-ALIGNED CROSS-MODAL POSITION ASSOCIATION

Although the position-aware conditioning paradigm effectively enables the learning of subject consistency from different references, it suffers from the training-inference gap in multi-subject generation. Typically, the text prompt is treated as a global condition, with a null positional coordinate (e.g., (0,0,0)). While this setup is sufficient to establish correspondence between the prompt and a single reference, its efficacy breaks down in multi-subject synthesis. At inference, when multiple references are present, this non-specific global guidance fails to properly ground textual semantics onto their respective visual concepts, causing attribute leakage and consistency degradation.

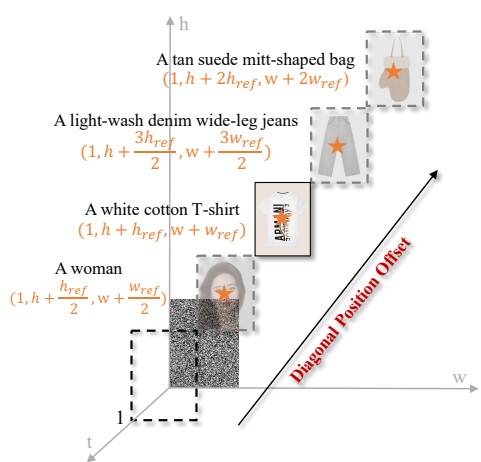

Figure 4: Illustration of the Center-Aligned Cross-Modal Position Association module.

To address this challenge of semantic grounding ambiguity, we introduce the Center-aligned Cross-modal Position Association (CCPA) module. The core idea is to move beyond undifferentiated, global textual-visual interaction and establish an explicit positional correspondence between textual descriptions and their intended visual concepts. As shown in Figure 4, we apply position embeddings not only to the noisy latent and reference image features, but also to the item descriptors associated with each visual concept. Basically, we encode positional information via 3D RoPE and separate the references and the target image via the offset in the first dimension, following Labs et al. (2025). Formally, given a position coordinate denoted by the triplet $u = (t, h, w)$, the position for noisy latent tokens is set as $u_t = (0, h, w)$. For the context tokens, the position coordinates have a basic spatial offset (h, w) and a virtual temporal offset 1. Combined with the pre-defined condition IDs, the complete position can be expressed by the following equation:

$$u_{ref}^i = (1, h + ID_i * h_{ref}, w + ID_i * w_{ref}), \quad i \in \{1, 2, 3, 4\}, \tag{2}$$

where $ID$ is condition index, $h$, $w$ and $h_{ref}$, $w_{ref}$ are height and width of the noisy target and context latents. As for the textual descriptors, we take the centroid coordinates of each visual concept's spatial location as their positional encoding, for an efficient and precise cross-modal alignment. With the same virtual temporal offset, position coordinates for item descriptions can be expressed as

$$u_{item}^i = (1, h + ID_i * h_{ref}/2, w + ID_i * w_{ref}/2), \quad i \in \{1, 2, 3, 4\}. \tag{3}$$

By introducing the centroid-aligned position coordinates, this module ensures that the semantic information from the prompt is routed exclusively to the pre-defined region of its corresponding visual concept, thereby enhancing visual consistency.

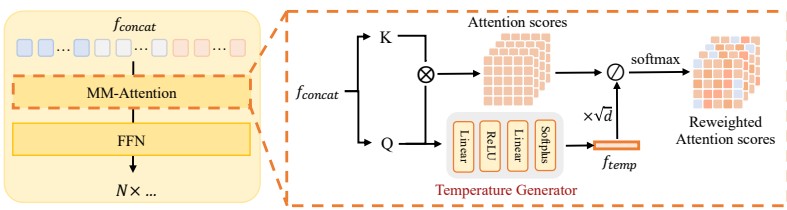

Figure 5: Illustration of the Dynamic Attention Modulation module.

## 3.4 DYNAMIC ATTENTION MODULATION

To relieve visual representation contention during model inference, a Dynamic Attention Modulation (DAM) module is proposed. Since the standard full-attention allows the noisy query to interact with all conditions indiscriminately, a contention for finite attention capacity can be triggered, compelling features of different subjects to compete for influence. Therefore, it is critical to dynamically modulate attention weights based on the query token of each timestep and network layer.

As shown in Figure 5, the concatenated multi-modal features $f_{concat}$ are processed through a series of $N$ MM-Attention blocks to facilitate feature interaction. To enable timestep-aware and layer-aware attention modulation, a Temperature Generator is introduced for the query tokens within each MM-Attention block of the DiT at every denoising timestep. To ensure stable and predictable behavior at the beginning of training, a specific initialization is set for the temperature generator. The generator consists of a simple Multi-Layer Perceptron (MLP) followed by a Softplus activation function. The final output temperature, $f_{temp}$, is calculated as

$$f_{temp} = \text{Softplus}(\text{MLP}(\text{W}_q(f_{concat})) + t_{min},$$ (4)

where $\text{W}_q$ is the parameter matrix for query projection and $t_{min}$ is a small constant (e.g., 0.01) to prevent the temperature from collapsing to zero. The predicted $f_{temp}$ is then applied as a token-wise scaling factor to the raw attention scores before the softmax operation, allowing the model to sharpen or soften its focus on specific tokens. This attention reweighting process can be written as

$$\mathcal{A} = \text{softmax}\left(\frac{QK^T}{\sqrt{d} \cdot f_{temp}}\right),$$ (5)

where $\mathcal{A}$ represents the reweighted attention scores after softmax, $d$ is the dimension of $Q$ and $K$.

Benefiting from this dynamic modulation process, the model performs an implicit feature routing for each query token, enhancing its focus on highly relevant features while simultaneously suppressing the influence of irrelevant ones. This mechanism efficiently optimizes the model's attention allocation, effectively mitigating the attention dilution problem inherent in multi-subject customization.

## 4 EXPERIMENTS

### 4.1 EXPERIMENTAL SETUPS

We adopt FLUX.1-Kontext-dev (Labs et al., 2025) as our backbone model, which has shown great single-reference generation capability. We finetune it with LoRA (Ryu, 2023) module of rank 512 exclusively on 4 categories (face/upper garment/lower garment/bag) of real-world, single-subject data pairs. The model is trained with a learning rate of 5e-6 and a total batch size of 16 on 16 NVIDIA H20 GPUs. The whole training process involves 30,000 iterations on single-subject data and the resolutions of synthesis and reference images are $1024 \times 768$ and $768 \times 576$, respectively. For inference, flow matching sampling with steps of 25 is adopted, with a guidance scale of 6.0.

**Evaluation data and metrics.** We collect a specialized test bench since there is no publicly available one for evaluating human-centric multi-subject customization. The face, clothing, and bag data are manually selected from both public datasets, e.g., DressCode (Morelli et al., 2022) and Deep-Fashion (Liu et al., 2016), and a small set of internal data to improve diversity. We compile 100 face-clothing pairs. For each pair, we generate 12 images of three representative human-centric scenes

Table 1: Quantitative comparison with S2I methods and ablations. Bold indicates the optimal result.

| Method | FaceSim | GME-I | GME-T | FashionCLIP-I | FashionCLIP-T | DINO |
|---|---|---|---|---|---|---|
| Kontext (Labs et al., 2025) | 0.482 | 0.632 | 0.706 | 0.617 | 0.349 | 0.374 |
| MS-Diffusion (Wang et al., 2024b) | 0.335 | 0.575 | 0.591 | 0.613 | 0.169 | 0.378 |
| UNO (Wu et al., 2025b) | 0.435 | 0.626 | **0.710** | 0.619 | 0.346 | 0.379 |
| DreamO (Mou et al., 2025) | 0.594 | 0.616 | 0.706 | 0.617 | 0.333 | 0.361 |
| OmniGen2 (Wu et al., 2025a) | 0.433 | 0.647 | 0.697 | 0.615 | 0.350 | 0.389 |
| Ours w/o CCPA | 0.633 | 0.641 | 0.693 | 0.634 | 0.347 | 0.394 |
| Ours w/o DAM | 0.619 | 0.656 | 0.698 | 0.638 | 0.349 | 0.401 |
| Ours | **0.678** | **0.665** | 0.699 | **0.642** | **0.351** | **0.405** |

(urban street, lifestyle indoor, minimalist studio) and four random seeds, yielding a total of 1,200 images for analysis. Following Wu et al. (2025b); Yuan et al. (2025), we evaluate the subject consistency and prompt following. For subject consistency, face similarity (FaceSim) (fal, 2024), GME (Zhang et al., 2024a) image score (GME-I), FashionCLIP (Chia, 2023) image score (FashionCLIP-I), and DINO (Oquab et al., 2023) score are measured. For prompt following, both GME (Zhang et al., 2024a) text score (GME-T) and FashionCLIP (Chia, 2023) text score (FashionCLIP-T) are evaluated. Benchmark and evaluation details are provided in Appendix C and Appendix D.

**Baselines.** We conduct a comprehensive set of both quantitative and qualitative experiments, benchmarking our method against the baseline model FLUX.1-Kontext-dev (Kontext) (Labs et al., 2025) and two categories of the state-of-the-arts. The first category is Subject-to-Image customization (S2I), which includes MS-Diffusion (Wang et al., 2024b), UNO (Wu et al., 2025b), DreamO (Mou et al., 2025), and OmniGen2 (Wu et al., 2025a). The second is controllable human image generation, including BootComp (Choi et al., 2025) and Parts2Whole (Fan et al., 2025). Implementation details for the baseline methods are provided in Appendix B.

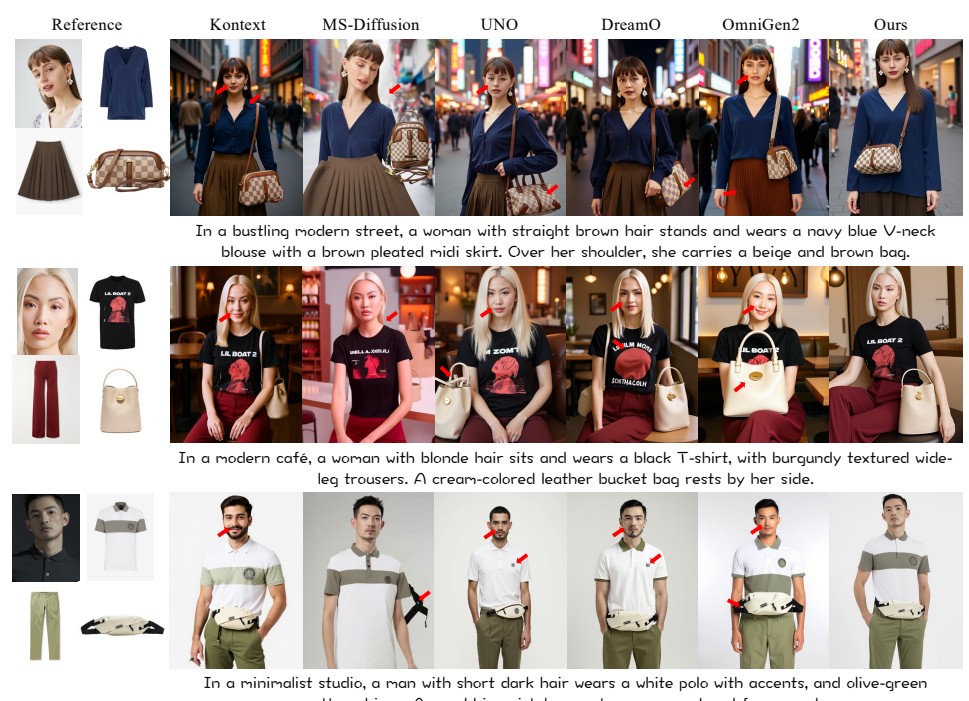

Figure 6: Qualitative comparison with state-of-the-art S2I methods.

Reference    BootComp    Ours    Reference    BootComp    Ours    Reference    BootComp    Ours

Figure 7: Qualitative comparison with BootComp (Choi et al., 2025).

Table 2: Quantitative comparison with controllable human image generation methods. Bold indicates the optimal result.

| Method | FaceSim | GME-I | FashionCLIP-I | DINO |
|---|---|---|---|---|
| Parts2Whole | 0.245 | 0.355 | 0.613 | 0.342 |
| BootComp | 0.100 | 0.401 | 0.555 | 0.328 |
| Ours | **0.670** | **0.669** | **0.649** | **0.418** |

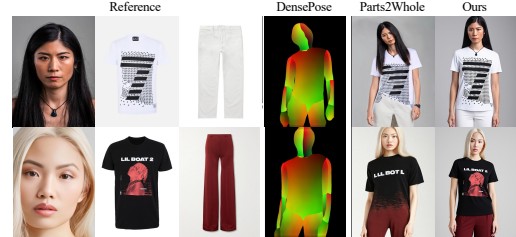

Reference    DensePose    Parts2Whole    Ours

Figure 8: Qualitative comparison with Parts2Whole (Fan et al., 2025).

## 4.2 COMPARISON WITH S2I METHODS

Qualitative comparison with S2I methods is shown in Figure 6. Results in three distinct outdoor and indoor settings are showcased. As can be seen, with a finite attention capacity, Kontext occasionally attends to only a subset of the reference images, exhibiting noticeable inconsistencies in facial features and clothing details. Results of MS-Diffusion are often blurry and show improper layering. Models trained on generated data, however, typically show overly smooth textures, especially for faces, which compromises their realism. Additionally, constrained by the upper limits of consistency in the training data, both UNO and DreamO exhibit clear shortcomings in maintaining clothing details. OmniGen2 suffers from poor facial consistency and implausible human-bag interactions. On the contrary, as shown in last column, our method demonstrates superior performance in multi-subject consistency, natural human poses, and high visual fidelity using only single-subject training examples. More qualitative results are shown in Appendix G.

Quantitative comparison is reported in Table 1. Our proposed method achieves the best overall performance compared to all baseline methods. In line with the qualitative results, our method demonstrates a substantial enhancement in face consistency, achieving a FaceSim score that is 0.084 higher than previous methods. Similar improvements are also observed across key image consistency metrics, i.e., GME-I, FashionCLIP-I, and DINO scores. Moreover, the superior performance of our method in generating consistent fashion-related attributes is verified by the highest FashionCLIP-T score. Crucially, this specialized improvement does not come at the cost of general prompt adherence, as evidenced by a GME-T score that remains competitive with the state-of-the-art methods.

## 4.3 COMPARISON WITH CONTROLLABLE HUMAN IMAGE GENERATION METHODS

Since BootComp and Parts2Whole have difficulty generating complex scenes, both the qualitative and quantitative comparison are conducted solely in simple background settings. Despite Boot-Comp's claim of face customization capability, generated images in Figure 7 often exhibit an absence of face areas or show negligible identity consistency (Column 3). An additional weakness observed is attribute leakage across reference subjects. For instance, the bag's strap and body colors bleed together in Column 1 and features from the bag are mistakenly applied to the pants in Column 2. For Parts2Whole, the comparison is conditioned on three references since it does not support a bag condition. Results in Figure 8 exhibit blurry boundaries between the upper and lower garments and copy-paste artifacts (Row 1). In contrast, our method achieves superior consistency, avoids attribute leakage, and generates natural interactions between multiple subjects.

Table 2 shows the quantitative comparisons. As can be seen, our method outperforms both Boot-Comp and Parts2Whole in all metrics, confirming the powerful advantage of our proposed training

Table 3: User study of the baseline methods. Bold indicates the optimal result.

| Method | Subject Consistency | Text Fidelity (subject) | Text Fidelity (background) | Visual Fidelity |
|---|---|---|---|---|
| Kontext (Labs et al., 2025) | 0.17 | 0.23 | 0.18 | 0.22 |
| MS-Diffusion (Wang et al., 2024b) | 0.04 | 0.04 | 0.03 | 0.02 |
| UNO (Wu et al., 2025b) | 0.09 | 0.12 | 0.20 | 0.19 |
| DreamO (Mou et al., 2025) | 0.10 | 0.09 | **0.22** | 0.17 |
| OmniGen2 (Wu et al., 2025a) | 0.19 | 0.18 | 0.17 | 0.10 |
| Ours | **0.41** | **0.34** | 0.20 | **0.30** |

Table 4: Quantitative ablations of the position-aware conditioning strategy. Bold indicates the optimal result.

| Method | FaceSim | GME-I | GME-T | FashionCLIP-I | FashionCLIP-T | DINO |
|---|---|---|---|---|---|---|
| Var.1 | 0.298 | 0.602 | 0.684 | 0.623 | 0.341 | 0.367 |
| Var.2 | 0.297 | 0.595 | 0.681 | 0.613 | 0.340 | 0.340 |
| Var.3 | **0.599** | **0.634** | **0.695** | **0.625** | **0.344** | **0.392** |

paradigm. By introducing the position-aware learning on single-subject data, our approach surpasses methods requiring specialized multi-subject training data for multi-reference human generation.

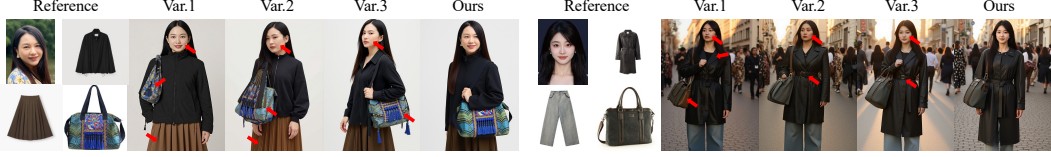

In a minimalist studio, a woman with long straight dark hair stands poised. She wears a black, matte-finish bomber jacket and a brown pleated midi skirt. Draped over her shoulder is a vibrant blue and gold patterned fabric bag adorned with tassels.

In a bustling urban street, a woman with long dark stands amidst the crowd. She wears a black leather trench and light-washed denim wide-leg. She carries a dark gray canvas and brown leather satchel.

Figure 9: Ablation comparisons of the position-aware conditioning strategy.

## 4.4 USER STUDY

To further demonstrate the superiority of our method, we conduct a user study, comparing it against the baseline methods. Specifically, we randomly sample 50 prompt-subject pairs from our benchmark for this comparison. 20 participants are asked to vote for the visual quality of the results generated by all methods. The assessment is performed across the following key dimensions: subject consistency, text fidelity concerning the subject, text fidelity concerning the background, and overall visual fidelity. As reported in Table 3, the results reveal that our method achieves the best overall alignment with user preferences. Despite a slightly higher text fidelity for the background, OmniGen2 shows noticeable deficiencies in subject-level consistency and overall image quality.

## 4.5 ABLATION STUDY

**Effect of the position-aware conditioning strategy.** To verify the effectiveness of the position-aware conditioning strategy, we implement three variants: (1) **Var.1** that uses index embeddings to distinguish different conditions, where all reference images and the noisy latent share the same positional coordinates, i.e., $u = (0, 0, 0)$; (2) **Var.2** that uses the virtual temporal dimension of 3D RoPE to distinguish different conditions, where the spatial offsets for all reference images are set to zero, i.e., $u = (t, 0, 0)$; (3) **Var. 3** that uses condition IDs to distinguish different reference subjects, i.e., Ours w/o CCPA and w/o DAM, for a comparison in identical experimental settings. The results of our full model are also provided. Results in Figure 9 and Table 4 clearly demonstrate that our

Reference w/o CCPA w/o DAM Ours  Reference w/o CCPA w/o DAM Ours

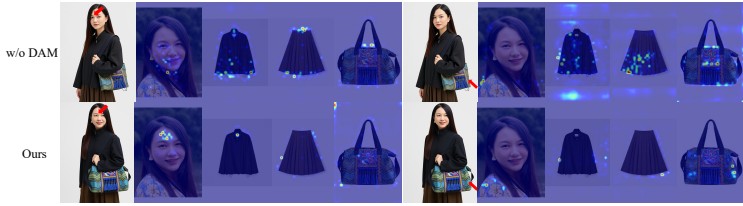

In a minimalist studio, a woman stands and wears a white cotton T-shirt, light-wash denim wide-leg jeans, and a tan suede mitt-shaped bag.

In a minimalist studio, a woman with long dark hair wears her white blouse with ruffled sleeves and plaid wool trousers. Draped over her shoulder is a beige crossbody bag with tan straps.

Figure 10: Qualitative comparisons of ablation studies of each proposed module.

Figure 11: Comparison visualizations of the Dynamic Attention Modulation module.

training strategy achieves superior overall effects, excelling particularly in subject consistency. This can be attributed to the strong spatial inductive bias introduced by our category-position binding, which offers a more concrete and distinguishable signal than optimization-reliant index embeddings and the subtle offset in the virtual temporal dimension.

**Effect of the CCPA module.** Figure 10 and Table 1 demonstrate results without the CCPA module. The lack of cross-modal information flow leads to a degradation in both text-to-image and image-to-image consistency, especially for complex texture details. This underscores the important role of cross-modal interaction and semantic grounding for conveying synergistic information.

**Effect of the DAM module.** As shown in Figure 10 and Table 1, the ablated model struggles to maintain overall consistency without the DAM. Its attention appears to be captured by prominent areas with rich textures, such as garments with complex patterns, allowing it to preserve their fidelity. However, this comes at the cost of neglecting other key regions like the face. Additionally, we provide an intuitive visualization in Figure 11. The DAM module effectively modulates the model's attention patterns for multiple references, compelling it to concentrate on the most relevant regions.

## 5 CONCLUSION

In this paper, we tackle the critical challenge posed by the cumbersome acquisition and often low-quality multi-subject training data in human-centric multi-subject customization. Our proposed method, One-for-All, introduces a novel position-aware conditioning strategy, enabling multi-subject consistency learning from only real-world, single-subject data. By integrating the Center-aligned Cross-modal Position Association and the Dynamic Attention Modulation module, our method successfully promotes semantic grounding and alleviates attention dilution. This, in turn, narrows the discrepancy between single-subject training and multi-subject inference, leading to a simultaneous enhancement in multi-subject consistency. Comprehensive experimental comparisons and analyses demonstrate the superiority of our method over both the state-of-the-art multi-subject customization and controllable human image generation methods. Ultimately, our method's robust multi-subject consistency showcases its potential for future scalability to accommodate more reference subjects, paving the way for richer and more controllable human visuals.

## ETHICS STATEMENT

Our model focuses on human-centric multi-subject customization, enabling the creation of high-fidelity images featuring specific reference subjects. This research is intended for beneficial and

creative real-world applications, such as hyper-realistic virtual try-on. We have ensured that all the data used during model training and evaluation are either publicly available or collected in strict compliance with all applicable ethical guidelines. However, we are aware that this research could potentially be misused for harmful purposes on social media, which may cause negative societal impacts. Therefore, we are committed to the responsible dissemination of our models as well as the resulting contents, limiting the release of our models and codes solely for research purposes. We also support the development of watermarking and detection methods to improve the traceability of synthetic contents. With proper ethical oversight, we believe our work will contribute meaningfully to progress in controllable image generation and unlock more creative application scenarios.

## REPRODUCIBILITY STATEMENT

Our work is committed to ensuring reproducibility. The implementation of our model is built upon publicly available codebases utilizing PyTorch. Essential hyperparameters and experimental configurations required to reproduce our results are described in Section 4.1. To facilitate further research, we will make our codebase, pre-trained model weights, and evaluation benchmark (detailed in Appendix C) publicly released upon acceptance of this paper.

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

# APPENDIX

## A  LLM USAGE STATEMENT

During the preparation of this manuscript, we utilize a Large Language Model (LLM) to assist with polishing the writing. The LLM is used for tasks such as improving grammar and refining sentences. The core research ideas, experimental designs, data analyses, and conclusions are exclusively conceived and formulated by the human authors.

## B  IMPLEMENTATION DETAILS FOR THE BASELINE METHODS

We provide the implementation details for the baseline methods in this section. For the evaluation of Kontext (Labs et al., 2025), since it does not support multi-reference inputs, we follow a common practice by concatenating the four reference images into a single large grid, serving as the model's input. Implementations of all comparative methods are based on their official open-source codes below:

- Kontext (Labs et al., 2025): `https://github.com/black-forest-labs/flux`;
- MS-Diffusion (Wang et al., 2024b): `https://github.com/MS-Diffusion/MS-Diffusion`;
- UNO (Wu et al., 2025b): `https://github.com/bytedance/UNO`;
- DreamO (Mou et al., 2025): `https://github.com/bytedance/DreamO`
- OmniGen2 (Wu et al., 2025a): `https://github.com/VectorSpaceLab/OmniGen2`;
- Parts2Whole (Fan et al., 2025): `https://github.com/huanngzh/Parts2Whole`;
- BootComp (Choi et al., 2025): `https://github.com/omnious/BootComp`.

## C  DETAILS OF HUMANBENCH

In this section, we provide details on the construction of our HumanBench, with data proportions provided in Table 5. To ensure diversity and comprehensive evaluation, we collect a total of 50 face images of individuals from different races. These images are sourced from both the DeepFashion (Liu et al., 2016) dataset and a small set of internal data. For the clothing categories, we gather 100 images in total, comprising upper garments, lower garments, and dresses, which are sourced from the DressCode (Morelli et al., 2022) dataset and internal data. Since there are no high-quality public datasets specifically for bags, we curate a collection of 20 bag images from internal sources. We ensure that all reference categories feature a well-balanced diversity, encompassing a wide range of human races, garment styles and textures, and bag types. Based on this collection, we construct our evaluation benchmark by randomly sampling and combining these reference types to create 100 distinct reference pairs. For the prompt preparation, we employ a two-stage process. First, a Large Language Model (LLM) is utilized to generate a concise item description for each reference subject. Then, these item descriptions are utilized to synthesize a global, detailed prompt. Concretely, this final prompt integrates the item descriptions into one of three representative human-centric scenes: urban street, lifestyle indoor, or minimalist studio background. An example of the prompt templates prepared in this stage is provided in Figure 12.

## D  DETAILS OF EVALUATION METRICS

To quantitatively evaluate model performance, we focus on two key aspects: subject consistency and prompt following. Evaluation details of these metrics are provided as follows.

- **FaceSim:** Measures the facial similarity between the generated image and the reference face image, evaluated using a pre-trained face recognition model (fal, 2024).

Table 5: Data proportions of HumanBench.

| Source | Face | Upper | Lower | Dress | Bag |
|---|---|---|---|---|---|
| DeepFashion (Liu et al., 2016) | 25 | 0 | 0 | 0 | 0 |
| DressCode (Morelli et al., 2022) | 0 | 43 | 34 | 10 | 0 |
| Internal | 25 | 7 | 6 | 0 | 20 |
| Total | 50 | 50 | 40 | 10 | 20 |

- **GME-I:** Assesses the average subject consistency between the generated and the reference subjects, based on the multi-modal model GME (Zhang et al., 2024a) and GroundingDINO (Liu et al., 2024).

- **FashionCLIP-I:** Evaluates the average subject consistency of the generated image with the reference subjects, based on FashionCLIP (Chia, 2023) and GroundingDINO (Liu et al., 2024).

- **DINO:** Measures the average subject consistency between the generated and the reference subjects, based on DINO (Oquab et al., 2023) and GroundingDINO (Liu et al., 2024).

- **GME-T:** Evaluates the prompt following between the generated image and the full prompt, based on the multi-modal model GME (Zhang et al., 2024a).

- **FashionCLIP-T:** Measures text-image consistency with a focus on fashion-related attributes, i.e., clothing descriptions, based on the FashionCLIP (Liu et al., 2016) model.

## E  BOARDER IMPACTS

Enabling human-centric multi-subject customization is a critical yet challenging frontier for T2I diffusion models, often hindered by the prohibitive cost of collecting paired training data for every possible subject combination. Our method directly tackles this challenge by achieving high-quality multi-subject synthesis from only single-subject training examples. This approach unlocks subject composition creativity and efficiently reduces data requirements, making human-centric multi-subject customization accessible for a wide range of applications. Ultimately, our method provides a critical foundation for future work, offering valuable insights to advance the field toward more scalable and flexible multi-subject customization.

## F  LIMITATIONS AND FUTURE WORK

While our method demonstrates high efficiency and capabilities in human-centric multi-subject customization, there still exist limitations. For instance, our approach focuses on customizing the human appearance, while the human pose and background generation relies solely on general prompt descriptions. This makes fine-grained control over the pose and background challenging. Therefore, for future work, we will actively explore extending the model to support explicit control over both the human pose and background. Moreover, we plan to integrate a broader range of human-centric conditions like shoes and accessories. This will lead to a promising avenue for more fine-grained human customization. With an expanded set of conditional inputs, it is natural that the proposed approach will unlock more creative and diverse application scenarios.

## G  MORE RESULTS

### G.1  MORE QUALITATIVE COMPARISONS

To further demonstrate the superiority of our method over existing approaches in human-centric customization, we provide additional visual comparisons in Figure 13. As can be seen, our method excels in simultaneous multi-subject consistency and detailed texture generation, even for complex clothing patterns in Row 3 and 6.

Stage 1

**Generate a short item description based on the following rules:**
# If the image shows a person's face: Describe the person's static physical appearance (e.g., gender, hair style, facial structure). Exclude any expressions, emotions, or head direction.
# If the image shows any other object (e.g., clothing, an accessory): Describe this item. Focus only on its physical appearance (e.g., color, material, style).
# No more than 10 words.
# Return a .json file with reference paths and item descriptions.
# Example 1: A man with short dark hair and round glasses.
# Example 2: A Vibrant blue canvas backpack with front pocket.

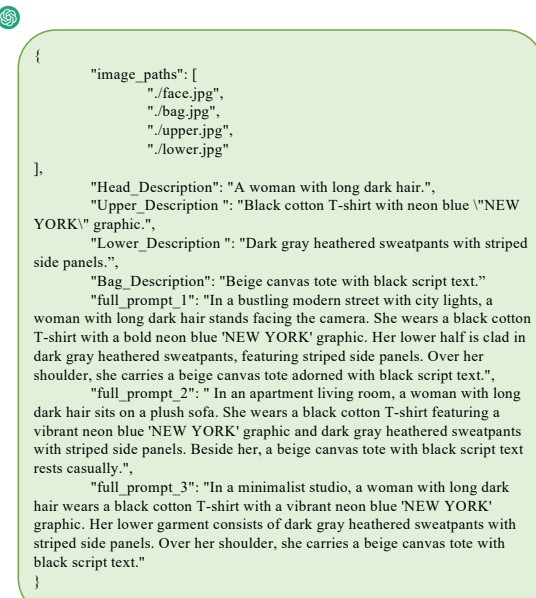

Stage 2

**Create a high-quality and descriptive prompt for generating an image of a person:**
# Combine the provided item descriptions and a selected scene from the options: [urban street, lifestyle indoor, minimalist studio].
# Return a .json file with reference paths, item descriptions, and full prompts.
# Example 1: In a bustling modern city street, a woman with long dark hair stands facing the camera. She wears a black cotton T-shirt with a bold neon blue 'NEW YORK' graphic. Her lower half is clad in dark gray heathered sweatpants, featuring striped side panels. Over her shoulder, she carries a beige canvas tote adorned with black script text.
# Example 2: In an apartment living room, a woman with long dark hair sits on a plush sofa. She wears a black cotton T-shirt featuring a vibrant neon blue 'NEW YORK' graphic and dark gray heathered sweatpants with striped side panels. Beside her, a beige canvas tote with black script text rests casually.

Figure 12: Examples of the two stages of prompt preparation.

## G.2 QUALITATIVE RESULTS WITH VARYING REFERENCES

In this section, we present qualitative results that demonstrate our model's capability to handle varying reference inputs. A key strength of our approach is its flexibility: trained on four reference types, the simple yet effective position-aware conditioning strategy allows our model accept any subset of these as conditions during inference. Specifically, Figure 14 showcases the synthetic results when conditioned on face identity, an upper garment, and a lower garment. Figure 15 demonstrates the results using only a face and an upper garment. Figure 16 demonstrates the results using only a face and a bag reference. Figure 17 presents the results conditioned on the upper and lower garment references, while Figure 18 presents the results for a face and a dress references. Notably, to handle the dress category, we supply the same dress image to the model twice: once as the reference for the upper garment and once for the lower garment.

## G.3 QUALITATIVE RESULTS UNDER VARIOUS SCENES

Figure 19 showcases our model's performance when conditioned on various reference subjects across diverse scenes, including indoor/outdoor settings and both natural and man-made landscapes. The model's ability to maintain high consistency across these varied scenes further demonstrates the effectiveness of our method for human-centric multi-subject customization and highlights its great potential for real-world applications.

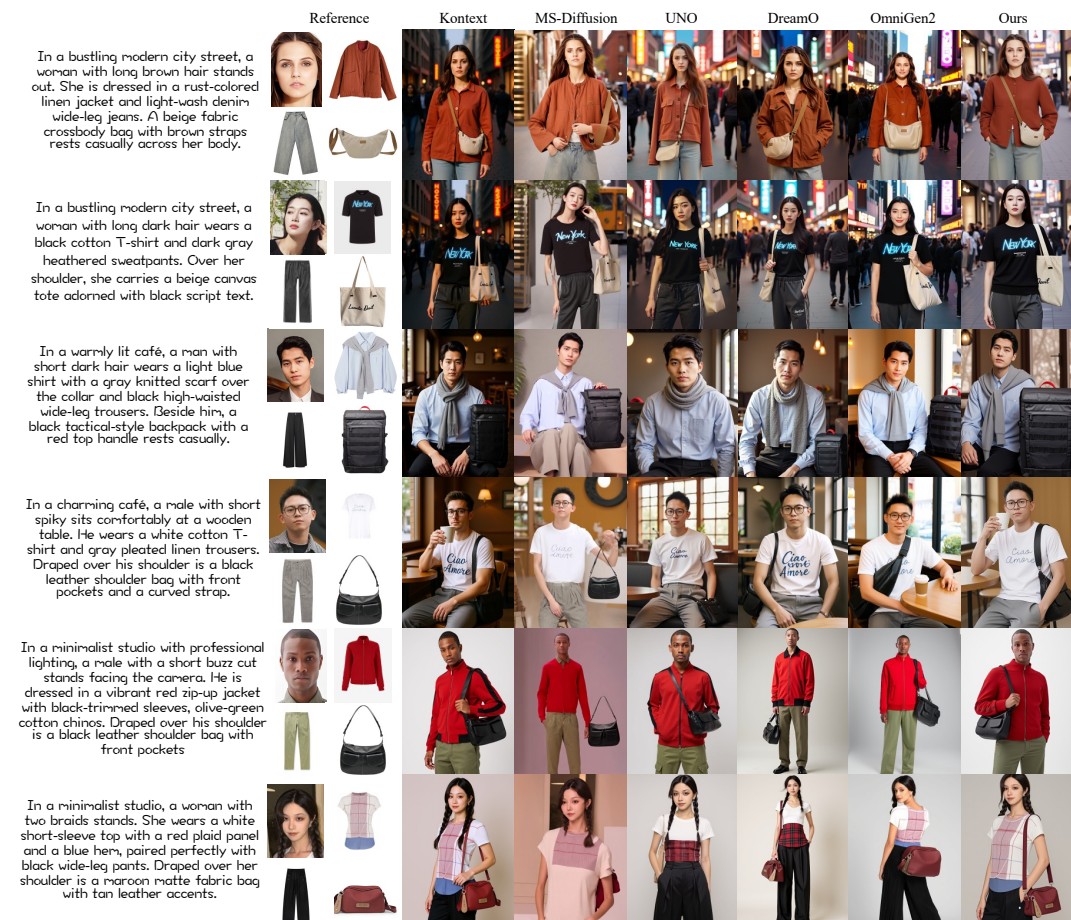

Figure 13: More qualitative comparisons with state-of-the-art methods.

## G.4 QUALITATIVE COMPARISONS WITH THE CLOSED-SOURCE MODELS

To benchmark our method against the closed-source models, we conduct qualitative comparisons with several lasted models, i.e., GPT-4o OpenAI (2025), Qwen-Image-Edit (v2509) Qwen (2025) and Nano Banana Google (2025). Comparison results are presented in Figure 20. While closed-source models often excel at rendering scenes with rich details, they frequently exhibit a degradation in subject consistency. This is evident in our comparisons, where GPT-4o struggles to capture realistic textures, Qwen-Image-Edit fails to preserve bag details, and Nano Banana struggles with facial identity preservation. On the contrary, our method demonstrates a consistent advantage in the core task of maintaining multi-subject consistency, especially for facial features.

## G.5 GENERALIZATION RESULTS

In this section, we evaluate the generalization performance of our method through a comprehensive set of experiments. Our evaluation is structured around three main scenarios: generalization on unseen cases, scalability to an increased number of subjects, and application to non-human subjects.

### G.5.1 GENERALIZATION ON UNSEEN CASES

**Results of face and scarf references.** To evaluate the model's generalization capabilities to other human-centric reference categories, we present its generation results using a face and a scarf as the reference. As shown in Figure 21, the model successfully composes the unseen scarf onto the person, demonstrating its robust baseline capability even when category-specific priors are not

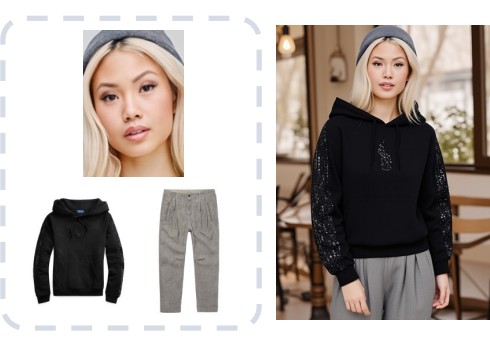 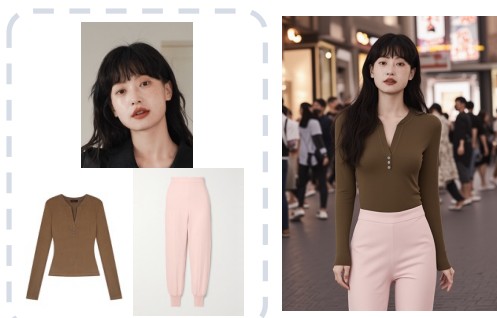

In a warmly lit café, a woman with blonde hair is dressed in a black hooded sweatshirt adorned with sequined, paired with gray pleated linen.

In a bustling modern city street with crowds, a woman with long dark stands. She wears a brown long-sleeve henley, paired with pale pink high-waisted tapered pants.

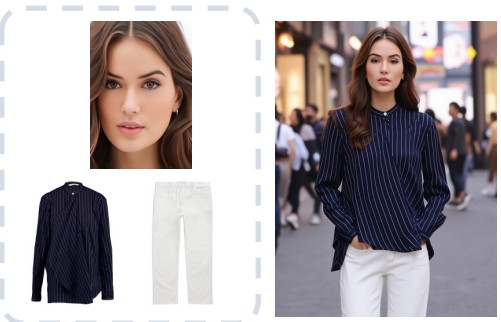 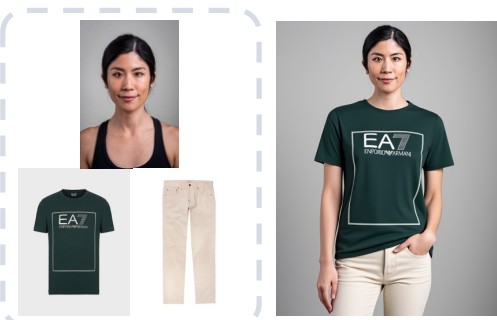

In a bustling modern city street with crowds, a woman with wavy brown faces the camera. She wears a navy blue pinstripe blouse and white cotton straight-leg pants.

In a minimalist studio with soft lighting, a woman with dark hair tied back stands. She wears a dark green cotton t-shirt, and slim-fit pale beige denim.

Figure 14: More qualitative results with face, upper, and lower garment references.

applied. It indicates that the model learns the broad concept of "wearable styles" of categories rather than overfitting to specific clothes like 'shirts'.

**Results of multi-top references.** Similarly, while we do not explicitly train for multiple same-category subjects, our framework exhibits a certain generalization for categories that share similar wearable interactions with the human body. For instance, to generate an image with two upper garments, we can provide a second 'upper garment' reference using the positional embedding originally designated for the 'lower garment'. The generation results are shown in Figure 22. As shown by the results, while the overall structure is synthesized correctly, there are losses of consistency in specific details, such as the collar and hem, which further validates that our category-position binding is essential for robust consistency.

**Results of cartoon character references.** Although our model is trained exclusively on photorealistic data, we find that it demonstrates a notable degree of generalization to the out-of-distribution cartoon characters. Generation results are presented in Figure 23 to showcase this capability, where the model successfully composes stylized subjects while preserving their core identity. This suggests our method's ability to generalize to unseen styles, indicating it learns a fundamental representation of subject identity and composition, rather than simply overfitting to the textures and style of the training domain.

### G.5.2 GENERALIZATION ON ADDITIONAL HUMAN-CENTRIC SUBJECTS

To investigate the model's scalability to more references, we conduct an extension experiment by collecting data for 'hat' and 'shoes' categories to create a challenging generation task with 5 and 6 reference subjects, respectively. The generation results with 5 and 6 reference subjects can be seen

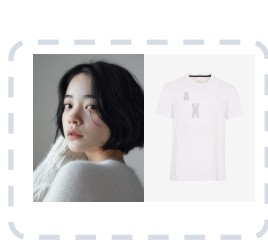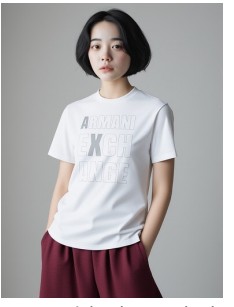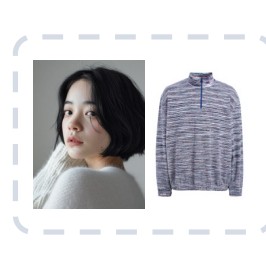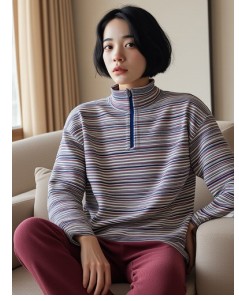

In a minimalist studio, a woman with short, dark wears a white cotton t-shirt with embossed gray. Her lower garment consists of burgundy textured wide-leg trousers.

In a warmly lit apartment living room, a woman with short dark sits comfortably on a sofa. She is dressed in a multicolored striped knit pullover, and burgundy textured wide-leg trousers.

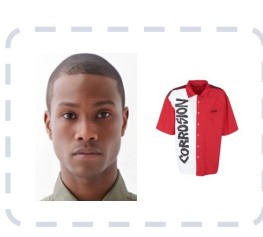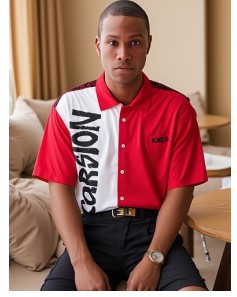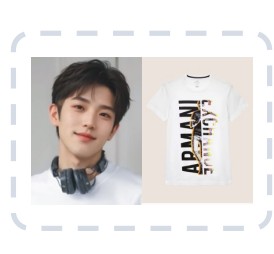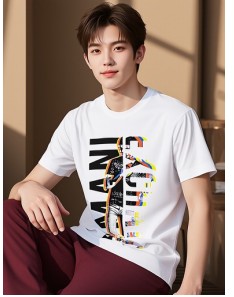

In a cozy café, a male with a short buzz cut wears a red and white short-sleeved, paired with black short pants

In a warmly lit café, a young male with short, wavy brown wears a white cotton T-shirt and burgundy textured wide-leg trousers.

Figure 15: More qualitative results with face and upper garment references.

in Figures 24 and 25. The generation results demonstrate that our method can successfully combine 6 reference subjects at once and achieve commendable performance.

Specifically, for 5 subjects involving a hat in Figure 24, the model demonstrates robust generalization across diverse hat materials and styles, including woolen beanies, sun hats, and felt hats. It accurately generates and composes these references while preserving the consistency of all other subjects. For the case of 6 subjects in Figure 25, which includes an additional hat and shoes, the generated images feature full-body standing portraits to better showcase the footwear. Similarly, despite being trained only on single-subject data, our method proves capable of modeling plausible interactions for subjects across a full-body composition. Furthermore, it can handles various shoe types, such as boots, flip-flops, and sneakers.

In summary, when we extend the current model to combine 6 subjects, there is no significant drop in generation quality. While some detail degradation is inevitable in full-body portraits due to the reduced screen percentage of each subject, the overall coherence and consistency remain strong. More importantly, all these commendable results rely solely on data-efficient, single-subject training, which further underscore the robust scalability of our framework.

### G.5.3 GENERALIZATION ON NON-HUMAN SUBJECTS

In this section, we conduct an extension experiment to test model generalization to non-human subjects. Specifically, we introduce a generic 'other' category (assigned to a condition ID: 5) and train the model on general single-subject data from the public Subject200k (Tan et al., 2024) dataset. Model performance is evaluated on the DreamBench (Ruiz et al., 2023) dataset. The generation results are presented in Figure 26. As shown, our model readily acquires the ability to generate these diverse non-human subjects with high consistency in various scenarios. This result strongly supports that our framework is domain-agnostic and robustly applicable beyond the primary human-centric focus.

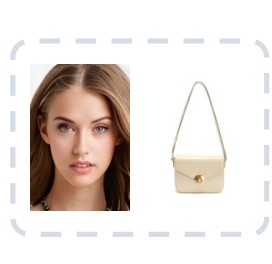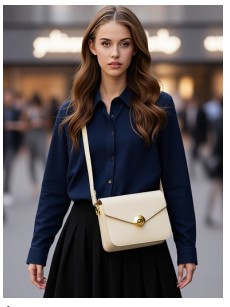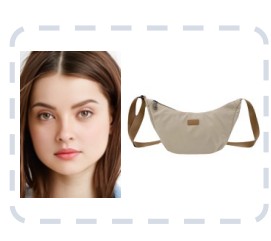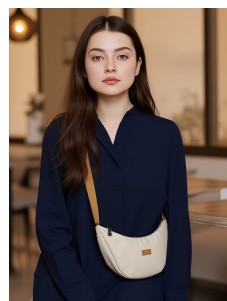

In a bustling city street, a young woman with long, wavy light brown hair is captured mid-stride. She is carrying a chic off-white shoulder bag with a sleek strap and a polished gold circular clasp.

In a modern café, a young woman with long brown hair sits. She wears a dark navy-blue long-sleeved dress, complemented by a beige crescent-shaped crossbody bag with a tan strap.

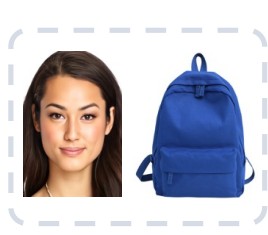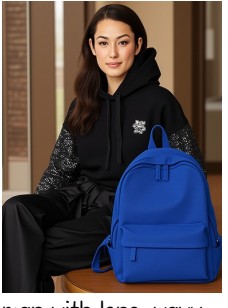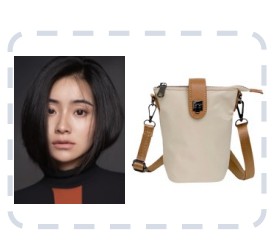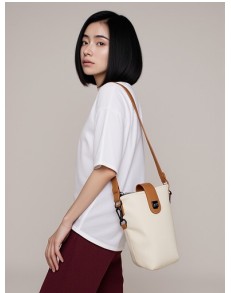

Outside a chic café, a woman with long, wavy hair sits comfortably. She is dressed in a black hoodie with sequined sleeves and is accompanied by a vibrant royal blue backpack resting beside her.

In a minimalist studio, a woman with a short black bob hair wears a simple white t-shirt and dark red trousers. Slung over her shoulder is a small, cream-colored bucket bag featuring a tan leather strap.

Figure 16: More qualitative results with face and bag references.

### G.6 COMPARISONS ON MORE COMMON CATEGORIES

To further validate the effectiveness of our proposed method, in this section, we conduct quantitative and qualitative comparisons on more common categories, i.e., inference without bags. To ensure a fairer comparison, the evaluation set here is composed solely of face images from the public DeepFashion (Liu et al., 2016) dataset and clothing images from the public DressCode (Morelli et al., 2022) dataset, comprising 50 image pairs across 3 different scenes for a total of 150 pairs. Consistent with the main experiments, we report the average results over 4 seeds. The qualitative and quantitative results are shown in Figure 27 and Table 6, respectively. The generated results in Figure 27 illustrate that even for in-distribution categories, previous methods still exhibit poor adherence to subject details and suffers from low visual fidelity. Meanwhile, the quantitative results in Table 6 consistently demonstrate the advantage of our method on these more common categories, especially for subject consistency.

Moreover, these findings suggest that the performance degradation in these previous methods stems from their failure to address the feature competition that arises from multiple reference inputs, rather than to limitations from not having seen a particular data category. Therefore, the strength of our method lies in its robustness in maintaining multi-subject consistency, and not just in the introduction of the new bag category.

## H DETAILS OF THE TRAINING DATA

Our training set consists of real-world, single-subject data pairs across four reference types: face, upper garment, lower garment, and bag, as illustrated by the green box examples in the middle of Figure 2. Specifically, the training data is sourced from e-commerce websites, featuring a rich

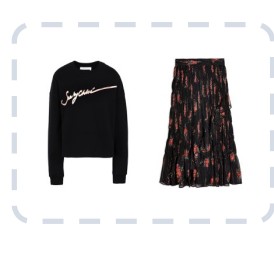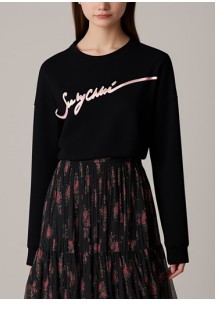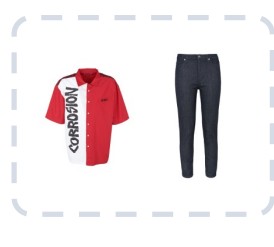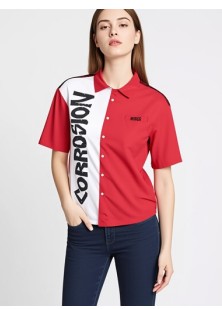

In a minimalist studio, a lady is dressed in a black sweatshirt with a script logo on the chest, and her lower half is clad in a black tiered skirt with a floral print.

In a minimalist studio, a lady stands facing the camera. She wears a striking red and white short-sleeved shirt, which boldly displays the word "CORROSION" across the chest. Her lower half is clad in dark blue denim straight-leg jeans, providing a classic contrast to the vibrant top.

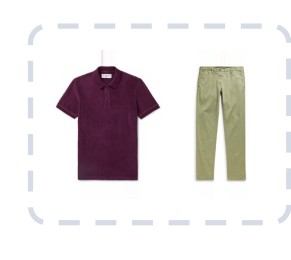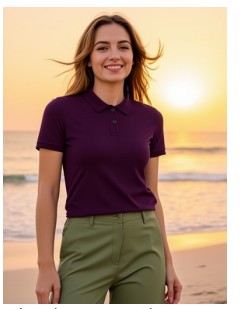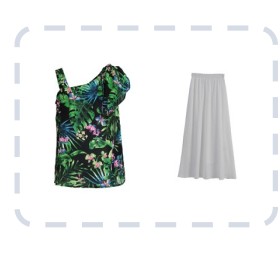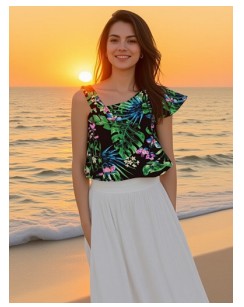

As the sun dips below the horizon, casting a warm golden glow, a lady stands on the wet sand of a beautiful beach. She is dressed in a deep purple textured polo shirt with short sleeves and a ribbed collar. Her olive-green cotton chinos with a straight fit complement her outfit perfectly.

As the sun dips below the horizon, casting a warm golden glow, a lady stands on the wet sand of a beautiful beach at sunset. She is dressed in a black tropical print one-shoulder blouse with ruffled sleeve. Her white textured cotton maxi skirt with elastic waistband flows gracefully around her.

Figure 17: More qualitative results with upper and lower garment references.

Table 6: Quantitative comparison with S2I methods on more common categories. Bold indicates the optimal result.

| Method | FaceSim | GME-I | GME-T | FashionCLIP-I | FashionCLIP-T | DINO |
|---|---|---|---|---|---|---|
| UNO (Wu et al., 2025b) | 0.528 | 0.718 | 0.699 | 0.700 | 0.304 | 0.501 |
| DreamO (Mou et al., 2025) | 0.694 | 0.693 | **0.705** | 0.658 | 0.306 | 0.454 |
| OmniGen2 (Wu et al., 2025a) | 0.538 | 0.709 | 0.700 | 0.704 | 0.303 | 0.483 |
| Ours | **0.732** | **0.736** | 0.696 | **0.739** | **0.312** | **0.523** |

diversity of subjects in various human-centric scenarios. The construction process begins by first gathering all available images for a given subject, followed by filtering them based on category label to retain only those that explicitly contain the subject. These filtered images are then formed into pairs, and a Vision-Language Model (VLM) is employed to verify that the image pairs depict the same subject, ensuring identity consistency. By repeating this process for each reference type, we collect image sets of the same subject from various viewpoints and scenarios. These collected images then undergo a series of preprocessing steps, i.e., cropping and segmentation, to obtain the final reference and target image pairs. Subsequently, the reference and target images are captioned respectively to produce the "item prompt" and "full prompt" required for model training. In total, the dataset comprises 40k pairs for face, 50k for upper garments, 40k for lower garments, and 100k for bags (with the larger volume for bags intended to account for their diverse carrying styles). To

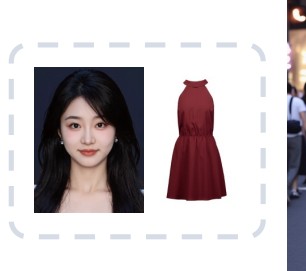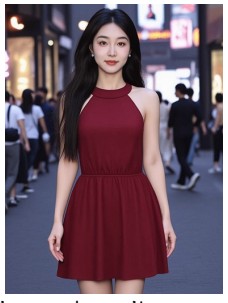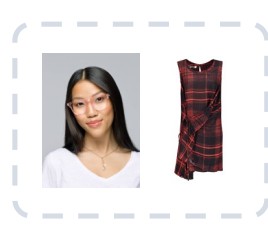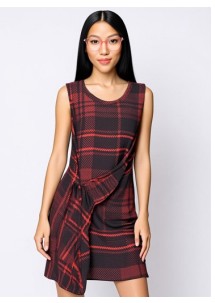

In the heart of a bustling modern city, a woman with long, dark hair faces the camera. She is dressed in a striking maroon halter-neck dress with an A-line skirt.

In a minimalist studio, a woman with long, straight dark hair and stylish pink glasses stands confidently facing the camera. She is elegantly dressed in a modern red and black plaid dress featuring an asymmetrical cut and a pleated side detail.

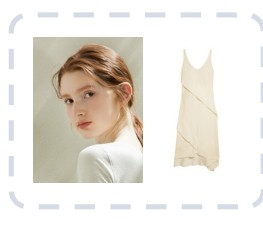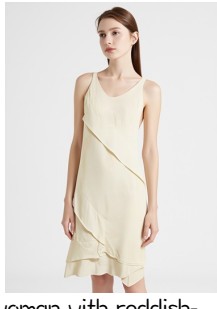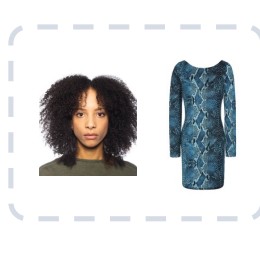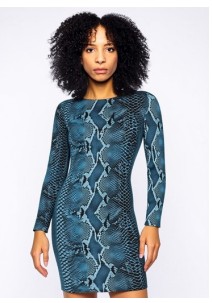

In a minimalist studio, a woman with reddish-brown hair stands, elegantly dressed in an off-white, layered, asymmetrical draped dress that flows gracefully around her.

In a minimalist studio, a woman with voluminous, curly dark afro hair stands confidently facing the camera. She is elegantly dressed in a long-sleeved, tight-fitting mini dress with a blue snakeskin print.

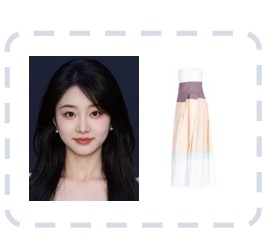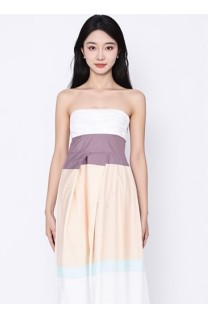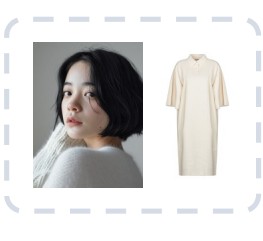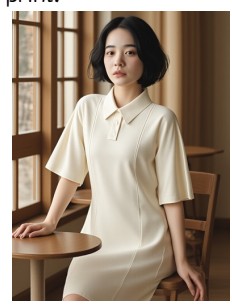

In a minimalist studio, a young Asian woman with long dark is dressed in a color-blocked strapless maxi dress, featuring a white bustier, a mauve mid-section, and a flowing skirt with panels of pale orange, light blue, and white.

In a warm-lit café, a woman with short, wavy black hair sits facing the camera. She is dressed in a cream-colored, short-sleeved, collared polo dress.

Figure 18: More qualitative results with face and dress references.

ensure data balance during training, a resampling ratio of 2:2:2:1 is employed for the face, upper-body, lower-body, and bag categories, respectively.

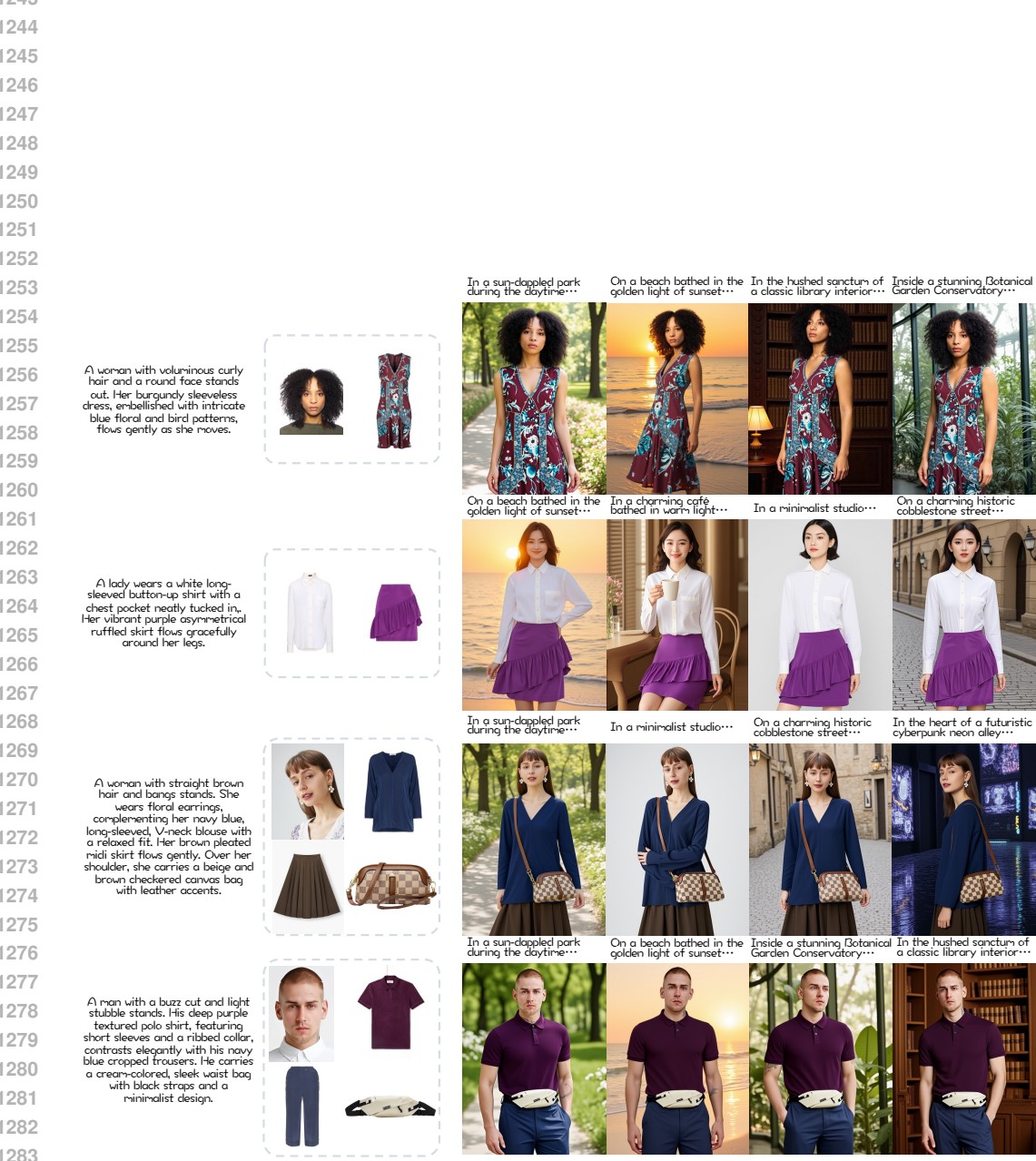

Figure 19: More qualitative results with different reference subjects under various scenes.

Reference     GPT-4o     Qwen-Image-Edit     Nano Banana     Ours

In a bustling modern street, a woman with straight brown hair stands and wears a navy blue V-neck blouse with a brown pleated midi skirt. Over her shoulder, she carries a beige and brown bag.

In a bustling modern city street, a woman with long dark hair wears a black cotton T-shirt and dark gray heathered sweatpants. Over her shoulder, she carries a beige canvas tote adorned with black script text.

In a modern café, a woman with blonde hair sits and wears a black T-shirt, with burgundy textured wide-leg trousers. A cream-colored leather bucket bag rests by her side.

In a minimalist studio, a man with short dark hair wears a white polo with accents, and olive-green cotton chinos. Around his waist, he sports a cream-colored fanny pack.

Figure 20: Qualitative comparisons with the closed-source models.

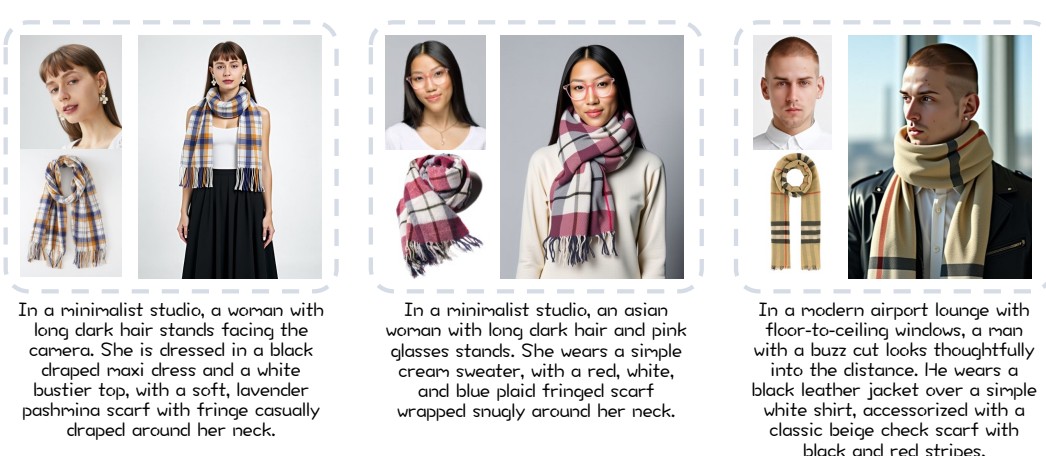

In a minimalist studio, a woman with long dark hair stands facing the camera. She is dressed in a black draped maxi dress and a white bustier top, with a soft, lavender pashmina scarf with fringe casually draped around her neck.

In a minimalist studio, an asian woman with long dark hair and pink glasses stands. She wears a simple cream sweater, with a red, white, and blue plaid fringed scarf wrapped snugly around her neck.

In a modern airport lounge with floor-to-ceiling windows, a man with a buzz cut looks thoughtfully into the distance. He wears a black leather jacket over a simple white shirt, accessorized with a classic beige check scarf with black and red stripes.

Figure 21: Qualitative results of face and scarf references.

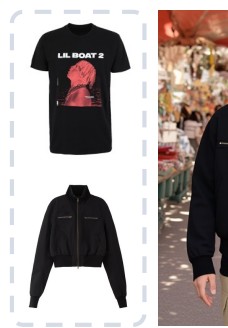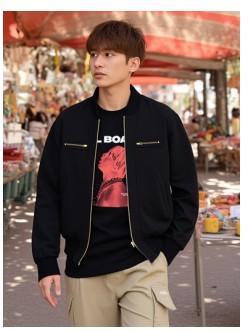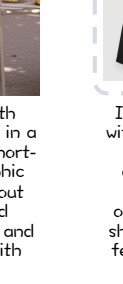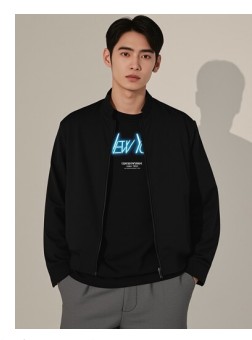

In an outdoor market, a young man with short, tousled brown hair stands, dressed in a layered ensemble. A glimpse of a black short-sleeve crew-neck t-shirt with a red graphic print is visible at the neckline, peeking out from under his unzipped black cropped bomber jacket adorned with gold zippers and ribbed cuffs. He pairs this stylish top with loose-fitting khaki cargo pants.

In a minimalist indoor setting, a young man with short, neatly styled dark hair stands in a relaxed pose. He is dressed in a layered outfit. A sleek black bomber jacket with a high collar and front zipper serves as the outer layer. Beneath it, a glimpse of a black short-sleeve T-shirt is visible at the neckline, featuring a neon blue "NEW YORK" graphic and small white text.

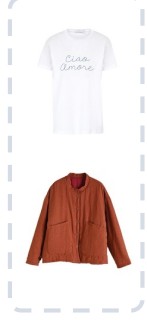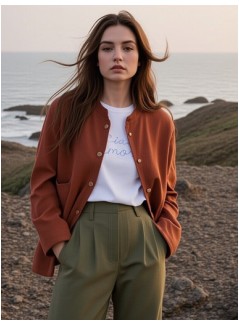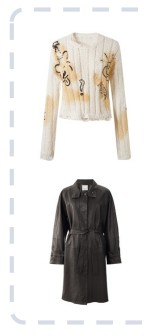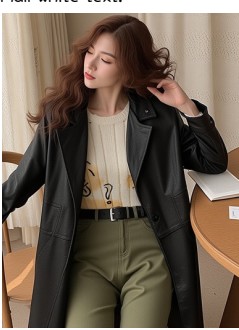

On a coastal cliffside, a young woman with long, wavy brown hair stands poised. She is dressed in a stylish layered ensemble. A rust-colored linen jacket with a mandarin collar and front buttons drapes elegantly over her shoulders. Beneath the jacket, a glimpse of a white short-sleeve T-shirt is visible at the neckline, where "Ciao Amore" script in light blue peeks out subtly. She pairs this with high-waisted olive green chinos.

In a quaint cafe, a young woman with wavy chestnut hair sits at a wooden table, dressed in a layered outfit. A glimpse of a cream and gold gradient knit sweater with abstract black embroidery is visible at the neckline, peeking out from under her unbuttoned black leather trench coat with a belted waist and notched collar. She pairs this with high-waisted olive green corduroy pants.

Figure 22: Qualitative results of multi-top references.

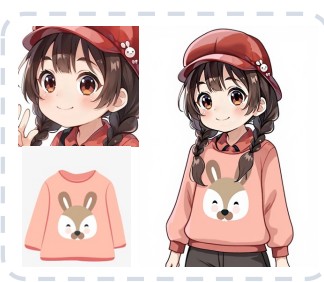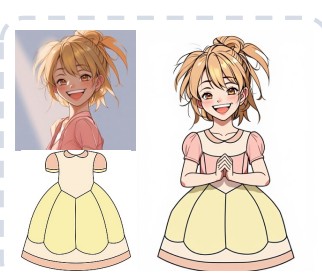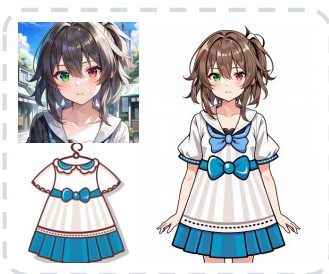

A full-body character design of an anime girl with brown pigtails and a red hat, standing against a solid white background. She is wearing a pink sweater adorned with a cute fawn graphic.

A full-body character design of an anime girl with a blonde messy bun and a big smile. She is wearing a lovely pastel yellow and pink gown with a Peter Pan collar.

A full-body character design of an anime girl with messy brown hair and heterochromia eyes (one green, one red). She is wearing a cute white and blue dress with a big bow at the waist and a Peter Pan collar.

Figure 23: Qualitative results of cartoon character references.

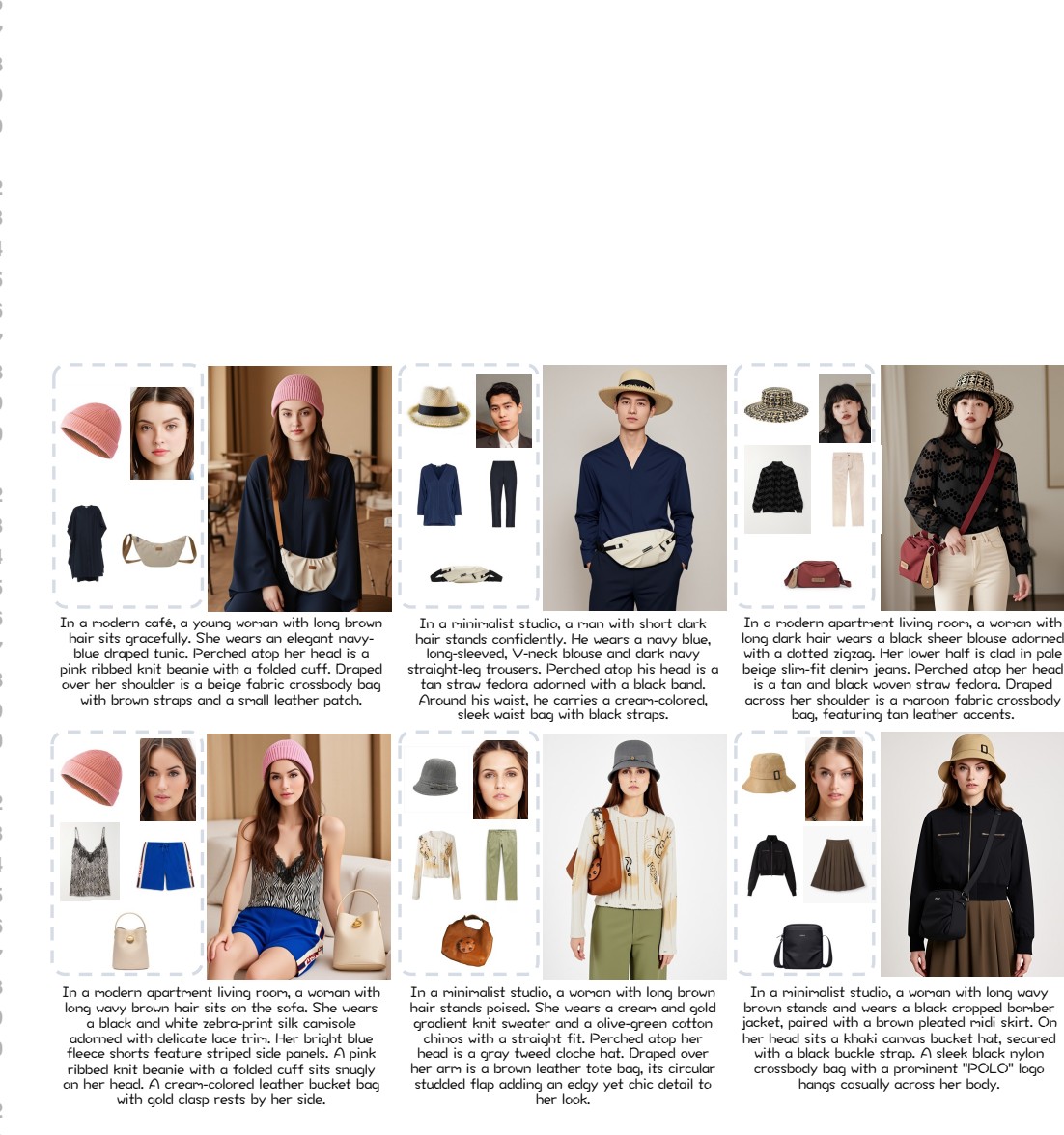

Figure 24: Qualitative results of additional hat references.

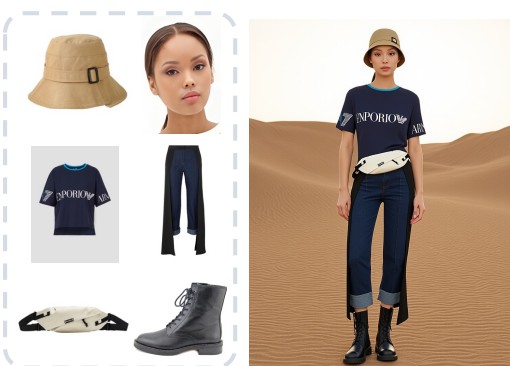 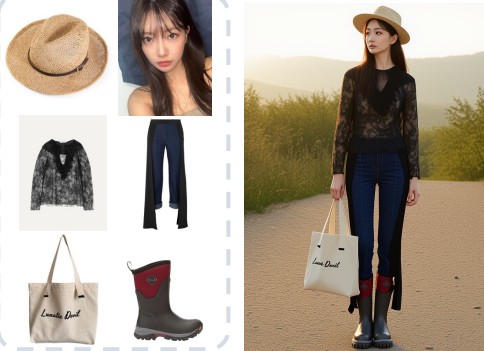

In the desert landscape, a woman with smooth, straight hairstyle stands confidently on a rugged desert trail. She wears a navy blue cotton T-shirt and blue denim jeans with black fabric panels and flared cuffs. On her head, she sports a khaki canvas bucket. Her feet are clad in sturdy black leather lace-up combat boots. Around her waist, she carries a cream-colored, sleek waist bag with black straps.

On a countryside path, a woman with long dark stands, her gaze directed towards the horizon. She wears a black lace blouse adorned with a sheer floral pattern and long sleeves. Her lower garment consists of blue denim jeans with black fabric panels and flared cuffs. A tan straw fedora with a brown leather band sits atop her head. Her feet are clad in dark gray rubber boots with red neoprene accents and a rugged tread. In her hand, she carries a beige canvas tote with black script text.

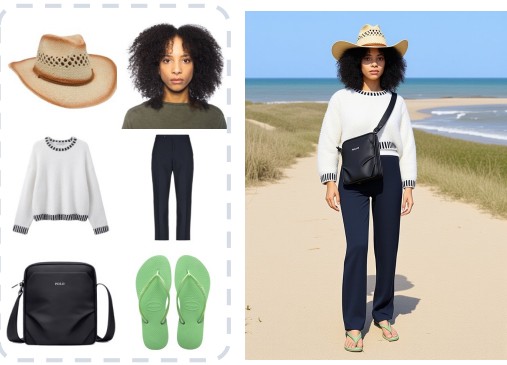 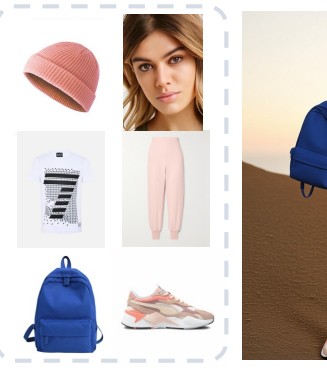

On a picturesque countryside, a woman with voluminous curly stands and wears a white knit sweater adorned with black and white striped trim that contrasts beautifully against her dark navy straight-leg trousers. Perched atop her head is a tan straw cowboy hat, featuring a brown band. Her feet are clad in pale green rubber flip-flops with braided straps. A sleek black nylon crossbody bag hangs across her shoulder, its geometric front panel gleaming subtly in the sunlight.

On a coastal cliff at golden hour, a woman with wavy blonde hair gazes out towards the horizon. She wears a white cotton t-shirt adorned with a black geometric print that contrasts beautifully against her pale pink high-waisted tapered pants. Her head is topped with a pink ribbed knit beanie. Her feet are clad in blush and coral mesh sneakers. A solid blue canvas backpack with a front pocket rests comfortably on her shoulders, completing her look.

Figure 25: Qualitative results of additional hat and shoes references.

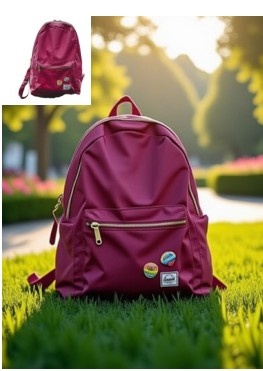
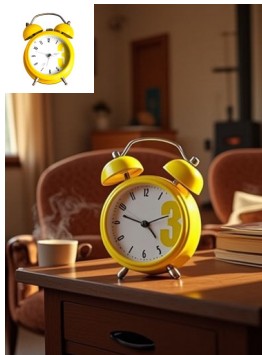
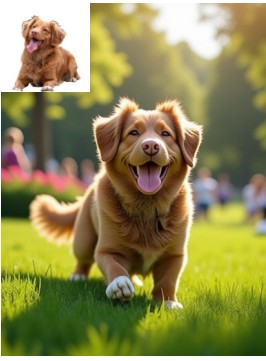

In the heart of a vibrant, lush park during the golden hours of daytime, a maroon nylon backpack with colorful pins and a front pocket rests casually on a soft patch of emerald grass.

In a warmly lit living room, the bright yellow retro alarm clock with a large number "3" and twin bells sits proudly on a wooden side table.

In the heart of a lush, green park during the daytime, a fluffy golden-brown dog with white-tipped paws and a joyful expression is frolicking through a soft carpet of emerald grass.

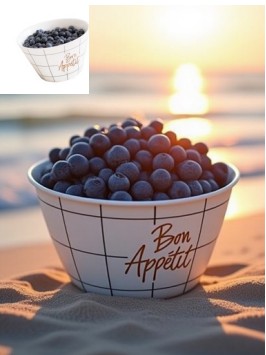
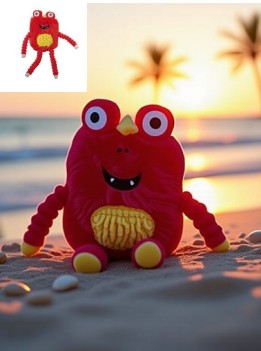
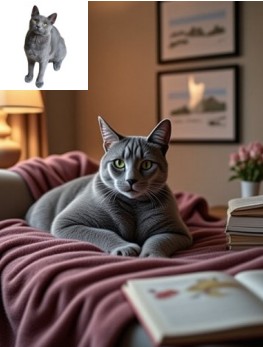

As the sun dips below the horizon, casting a warm, golden glow across the serene beach, a white ceramic bowl with a grid pattern and gold "Bon Appétit" text sits elegantly on a soft, sandy patch.

As the sun dips below the horizon, casting a warm, golden glow across the serene beach, a red plush toy with white eyes, yellow beak and feet, and a yellow belly sits contentedly on a soft, sandy patch.

In a warmly lit living room adorned with soft furnishings and tasteful decor, a gray short-haired cat with striking green eyes lounges on a plush pink fabric draped over the arm of a vintage sofa.

Figure 26: Qualitative results of non-human subjects.

Figure 27: Qualitative comparisons on more common categories, i.e., face and clothing references.

