# OpenReview forum: "One-for-All: Towrads Human-Centric Multi-Subject Customization from Single-Subject Examples"
_ICLR.cc/2026/Conference — Submitted to ICLR 2026_

### Official Review · Reviewer_iKbq · 2025-10-26

**Soundness:** 3
**Presentation:** 3
**Contribution:** 2
**Rating:** 4
**Confidence:** 4

**Summary:**

To address the challenge of acquiring high-quality multi-agent data for human-centric multi-agent customization, this paper proposes a One-for-All framework that establishes multi-agent consistency by learning solely from real-world single-agent examples.
The proposed framework introduces a Center-Aligned Cross-Modal Position Association (CCPA) module to facilitate interaction between visual references and textual descriptions, and incorporates a Dynamic Attention Modulation (DAM) mechanism to adaptively adjust attention weights, effectively mitigating issues of semantic grounding ambiguity and attention dilution.
Extensive experiments, including evaluations on HumanBench, demonstrate that One-for-All, even when trained exclusively on single-agent data, surpasses all baseline methods in multi-agent consistency, natural human pose generation, and visual fidelity, while exhibiting strong generalization capabilities.

**Strengths:**

1. This paper is well-written and presented.

2. The paper proposes One-for-All, a new framework that achieves multi-subject consistency using only real-world single-subject data. Through category-specific position conditioning, the model assigns unique conditional IDs and enforces position-aware learning, enabling robust multi-subject generation without relying on costly multi-subject datasets.

3. The Center-Aligned Cross-Modal Position Association (CCPA) module resolves semantic grounding ambiguity by explicitly linking textual descriptions to corresponding visual concepts through 3D RoPE-based position encoding. This centroid-aligned design ensures accurate text–image correspondence and enhances overall visual consistency.

4. The Dynamic Attention Modulation (DAM) mechanism alleviates feature competition among multiple subjects by dynamically adjusting token-wise attention weights through a learnable temperature generator. This adaptive control sharpens attention on relevant features and suppresses interference, improving multi-subject coherence and fidelity.

**Weaknesses:**

1. The proposed CCPA module lacks clear novelty, as similar position re-encoding strategies have been explored in prior works such as OmniControl, OmniControl-2, and EasyControl. The idea of using positional embeddings to handle multi-object reference conflicts is already well-established, making this component less distinctive in contribution.

2. The DAM mechanism appears relatively generic, relying solely on adaptive attention modulation without introducing explicit multi-object constraints or losses. It remains unclear whether such adaptive weighting alone can effectively mitigate feature competition among multiple subjects, and further analysis or ablation would be needed to validate its actual impact.

**Questions:**

1. How does the proposed CCPA module differ from prior works such as OmniControl, OmniControl-2, and EasyControl, which also employ position re-encoding to resolve multi-object reference conflicts? It would be helpful to clarify what specific innovation or design improvement CCPA introduces beyond these established approaches.

2. The DAM mechanism relies on adaptive attention scaling to alleviate feature competition, but it is unclear whether such self-learned modulation is sufficient to handle multi-object conflicts, especially when multiple subjects occupy overlapping spatial regions. How does the model ensure effective disentanglement or balance of attention in such cases without explicit multi-object constraints or loss terms?

---

> ### Author Response · Authors · 2025-11-24
> **Response (1/1)**
>
> **Q1:** How does the proposed CCPA module differ from prior works such as OmniControl, OmniControl-2, and EasyControl, which also employ position re-encoding to resolve multi-object reference conflicts? It would be helpful to clarify what specific innovation or design improvement CCPA introduces beyond these established approaches.
>
> **A1:** We sincerely appreciate the reviewer for pointing out this. Our work is indeed motivated by a specific limitation in prior work, which we would like to articulate as follows:
>
> **Limitation of Existing Methods:** We acknowledge that methods like OmniControl, OmniControl-2, and EasyControl use spatial position offsets to differentiate multiple subjects. However, they assign a static position index (e.g., (0,0)) to the text condition. This implementation, while enabling a global conditioning effect, sacrifices directional interaction with specific visual features from reference images.
>
> **Resulting Challenge:** This lack of targeted cross-modal interaction poses an obvious bottleneck, especially in multi-subject customization, where global-only textual guidance fails to properly ground textual semantics onto their respective visual concepts, causing attribute leakage and consistency degradation.
>
> **Our Contribution (CCPA):** To address this, our method introduces the CCPA module, which introduces **an explicit positional alignment between same-category subjects in the text and image conditions**. This fosters a cross-modal synergy, where textual information and visual information work in concert, rather than in isolation or even competition, to boost generation consistency. By enabling this targeted, collaborative conditioning, our approach ensures a more robust guarantee of multi-subject consistency. The effectiveness of this design is validated by our ablation study, presented in **Figure 10 and Table 1**.
> Thank you again for allowing us to elaborate on this point.
>
> **Q2:** The DAM mechanism appears relatively generic, relying solely on adaptive attention modulation without introducing explicit multi-object constraints or losses. It remains unclear whether such adaptive weighting alone can effectively mitigate feature competition among multiple subjects, and further analysis or ablation would be needed to validate its actual impact.
>
> **A2:** Thank you for this insightful comment, and we apologize for the possible ambiguity. We would like to clarify why the DAM's design is not only sufficient but also intentionally elegant for this task.
> **It's correct that the DAM does not rely on explicit multi-object constraints. This is a deliberate design choice aligned with our central contribution: learning multi-subject consistency from single-subject data.** Introducing an explicit multi-object loss would necessitate multi-subject training pairs, thereby undermining our primary goal of reducing data construction complexity. However, this simplicity does not come at the cost of subject consistency. On the contrary, through the DAM, we achieve a more elegant disentanglement that prevents the entanglement of multi-reference features.
>
> The key to DAM's effectiveness lies in how it resolves ambiguity and conflicts when standard attention is insufficient. In high-conflict cases where a query token finds multiple reference features almost equally relevant (e.g., multiple subjects occupy overlapping regions), a standard attention mechanism would produce blended, ambiguous weights. Our DAM is proposed to resolve this failure. **As a query-based modulator, it learns to recognize these high-conflict contexts from each query token.** In response, it predicts a very low reweighting score. This low temperature is then the crucial component for disentanglement. It is applied uniformly to all relevance attention weights for that query, forcing the softmax function into a "winner-takes-all" state. **This mathematically amplifies even minor differences in the initial scores, transforming a hesitant weight distribution into a decisive one and effectively selecting a single subject's features.** This entire process, both qualitatively and quantitatively validated by our ablation study and attention visualizations (**Figures 10 and 11, Table 1**), is how our seemingly "generic" module robustly generalizes from single-subject training to solve complex multi-subject inference scenarios without explicit constraints or complex optimization.

---

### Official Review · Reviewer_fHmp · 2025-10-28

**Soundness:** 2
**Presentation:** 2
**Contribution:** 2
**Rating:** 4
**Confidence:** 4

**Summary:**

This paper introduces Onefor-All, a framework that pioneers a new paradigm by learning multi-subject consistency from only real-world, single-subject examples, breaking the dependency on curated multi-subject data. Building upon this, it unlocks the full potential of this paradigm shift by introducing two key designs that ensure robust multisubject consistency.

**Strengths:**

1. This paper presents a novel framework that learns robust multi-subject consistency by leveraging only real-world, single-subject data.
2. Two novel mechanisms are proposed to adapt to paradigm shift and ensure multisubject consistency.
3. Extensive comparisons with state-of-the-art methods on our HumanBench demonstrate the superiority of the proposed method.

**Weaknesses:**

1. Can you show some attention maps or other visualizations to prove that it's correctly 'routing' the features for each subject to the right place in the image?
2. What happens when you try to combine even more subjects, like 6 or 8 at once? Does the quality start to drop?
3. How does your model learn realistic interactions, like how a shirt wrinkles when a person wears it, or how a bag strap presses into a jacket?
4. How generalizable is the method? For example, if we change the scene to cartoon characters, will it still work?
5. In your comparison images, it would be super helpful if you could add arrows or circles to point out exactly where your method is better.
6. Have you compared your results to the closed-source models?

**Questions:**

Please see the weakness.

---

> ### Author Response · Authors · 2025-11-24
> **Response (1/2)**
>
> **Q1:** Can you show some attention maps or other visualizations to prove that it's correctly 'routing' the features for each subject to the right place in the image?
>
> **A1:** Thank you for this valuable comment. Visualizing the attention flow is indeed a crucial validation for the DAM module, and we have provided such a visualization analysis to demonstrate its effectiveness in **Figure 11**. **As shown in the attention visualizations, the DAM's query-based dynamic relevance re-weighting effectively concentrates the model's attention onto the most relevant reference regions for any target query.** This corrects cases where attention would otherwise be scattered or incorrectly focused on irrelevant objects. That is, **correctly 'routing' the features for each subject to the right place in the image**, validating the description "implicit feature routing" in Section 3.4 of the main manuscript - "Benefiting from this dynamic modulation process, the model performs an implicit feature routing for each query token, enhancing its focus on highly relevant features while simultaneously suppressing the influence of irrelevant ones."
>
> **Q2:** What happens when you try to combine even more subjects, like 6 or 8 at once? Does the quality start to drop?
>
> **A2:** Thank you for this excellent and forward-looking comment about the scalability of our method. To investigate this, we conduct an extension experiment by collecting data for 'hat' and 'shoes' categories to create a challenging generation task with 5 and 6 reference subjects, respectively. The generation results with 5 and 6 reference subjects are depicted in **Figures 24 and 25 of Appendix G.5.2**, respectively. **The experimental results demonstrate that our method can successfully generalize to 6 reference subjects at once and achieve commendable performance.**
>
> Specifically, for 5 subjects involving a hat, the model demonstrates **robust generalization across diverse hat materials and styles**, including woolen beanies, sun hats, and felt hats. It accurately generates and composes these references while preserving the consistency of all other subjects.
> For the case of 6 subjects, which includes an additional hat and shoes, the generated images feature full-body standing portraits to better showcase the footwear. Similarly, **despite being trained only on single-subject data, our method proves capable of modeling plausible interactions for subjects across a full-body composition**. Furthermore, it effectively handles various shoe types, such as boots, flip-flops, and sneakers.
>
> IIn summary, **when we extend the current model to combine 6 subjects, there is no significant drop in generation quality**. While some detail degradation is inevitable in full-body portraits due to the reduced screen percentage of each subject, the overall coherence and consistency remain strong. More importantly, all these commendable results rely solely on data-efficient, single-subject training, which further underscore the robust scalability of our framework.
>
> **Q3:** How does your model learn realistic interactions, like how a shirt wrinkles when a person wears it, or how a bag strap presses into a jacket?
>
> **A3:**
> Thank you for this important comment. **The ability of our model to generate such realistic interactions is learned through a powerful combination of implicit data-driven learning and explicit strategy-driven guidance.**
>
> Basically, on the data side, the target images in our single-subject training data are real-world photographs that naturally exhibit these realistic wearable interactions. For instance, they serve as rich examples of how fabrics wrinkle and drape over a human body, or how bag straps create pressure and occlusion on clothing. As the model is trained to reconstruct these authentic target images, it learns to render these subtle yet crucial interaction details. This is highly practical, since human-centric objects always appear in the context of a human wearer.
> Furthermore, and more importantly, this implicit learning is enhanced by our position-aware conditioning strategy. The binding between clothing categories and their positional embeddings does more than just place subjects: it facilitates the learning of specific, category-related wearing priors. This is further evidenced by our generation results with an unseen scarf reference and multiple top garments, as illustrated in Figures 21 and 22 of Appendix G.5.1.

---

> ### Author Response · Authors · 2025-11-24
> **Response (2/2)**
>
> **Q4:** How generalizable is the method? For example, if we change the scene to cartoon characters, will it still work?
>
> **A4:** Thank you for this insightful comment regarding the model's generalizability to the out-of-distribution styles. **Although our model is trained exclusively on photorealistic data, we find that it demonstrates a notable degree of generalization to the out-of-distribution cartoon characters.** Some generation results are presented in **Figure 23 in Section G.5.1 of the Appendix** to showcase this capability, where the model successfully composes stylized subjects while preserving their core identity.
> This suggests that our method learns a more fundamental understanding of subject identity and composition, rather than simply overfitting to the textures and style of the training domain. The subtle losses in consistency of details can likely be addressed by supplementing with a small amount of data in the corresponding style.
>
>
> **Q5:** In your comparison images, it would be super helpful if you could add arrows or circles to point out exactly where your method is better.
>
> **A5:** Thank you for this helpful suggestion. We agree that visual annotations help improve the clarity of the comparisons. Following the advice, we have updated Figures 6, 9, and 10 with arrows to pinpoint the key areas of improvement. Please refer to the revised manuscript for details.
>
> **Q6:** Have you compared your results to the closed-source models?
>
> **A6:** Thank you for this excellent suggestion to benchmark our method against the closed-source models. We conduct qualitative comparisons with several lasted models, i.e., GPT-4o, Qwen-Image-Edit and Nano Banana.
> Comparison results are presented in **Figure 20 in Section G.4 of the Appendix**. While closed-source models often excel at rendering scenes with rich details, they frequently exhibit a degradation in subject consistency. This is evident in our comparisons, where GPT-4o struggles to capture realistic textures, Qwen-Image-Edit fails to preserve bag details and Nano Banana struggles with facial identity. On the contrary, **our method demonstrates a consistent advantage in the core task of maintaining subject consistency, especially for facial features**.
> Crucially, the value of our framework is best understood by considering the specific human-centric domain. While leading closed-source models are optimized for general-purpose scenarios, they often struggle with human-centric consistent generation, which is actually highly-demanded. Our method, in contrast, is carefully designed to address this challenge. **It focuses on solving the critical problem of human-centric consistency, while operating with significantly lower data requirements.** In summary, our work provides a new solution for a challenging problem that general-aimed closed-source models are not designed to solve.

---

### Official Review · Reviewer_7JcY · 2025-10-30

**Soundness:** 3
**Presentation:** 3
**Contribution:** 2
**Rating:** 6
**Confidence:** 4

**Summary:**

This paper proposes One-for-All, a novel framework designed to address the challenge of human-centric multi-subject customization in image synthesis without relying on curated multi-subject data. It solves the problem of subject inconsistency by learning multi-subject consistency from single-subject real-world examples. Firstly, the paper introduces a Center-aligned Cross-Modal Position Association (CCPA) module, which aligns visual references with their textual descriptions by using center-based positional encoding. This design enhances semantic grounding across modalities and strengthens intra-subject consistency. Secondly, the authors propose a Dynamic Attention Modulation (DAM) mechanism that predicts token-wise attention weights to counteract attention dilution from multiple subjects. This ensures that critical features receive sufficient focus and preserves multi-subject consistency. Finally, extensive experiments on the proposed HumanBench demonstrate that the effectiveness of the proposed methods.

**Strengths:**

1. The task is interesting.
2. The motivation somewhat makes sense.
3. The proposed method is simple yet effective.
4. The framework learns multi-subject consistency from single-subject data, which is much easier to obtain.

**Weaknesses:**

1. Limited Input Scalability. The proposed framework supports only four input types (face, upper garment, lower garment, and bag), with the restriction that each type can appear only once and must be encoded using its corresponding positional embedding. This design limits the method's scalability, making it unsuitable for scenarios involving more diverse subjects, such as multiple faces, multiple tops, several bags, or additional accessories.
2. The paper repeatedly refers to a small set of internal data, yet the scale and details of this dataset are not disclosed. This lack of transparency makes it difficult for other researchers to reproduce the results or conduct fair comparisons.

**Questions:**

**Major Comments**
1. Is the proposed CAM module trained only on the single-subject dataset? From Fig. 5, it seems that the input is limited to Q. How does the module infer the required weights solely from the target image? Furthermore, how does it generalize to a larger number of reference items? Are the re-weighting scores computed independently for each item?
2. Please provide more details on the internal dataset used in the paper. Clarifying its specific characteristics would benefit the community as a whole.
3. To strengthen the argument, it would be more convincing to include objective results that validate the effect of the position-aware conditioning strategy. I may have overlooked it, but I only found visual comparisons in Figure 9.

**Minor Comments**
1. The visual comparison figures  (e.g.,  Fig. 6, Fig. 9, Fig. 10) do not display the corresponding input text of the generated images.
2. In several visual comparisons (e.g., Fig. 9, Fig. 10), the differences between methods are not very clear. I suggest highlighting the key differences directly in the figures for better readability.
3. The proposed human-centric multi-subject customization task is very closely related to the multi-garment virtual try-on task. Therefore, it would be beneficial to discuss more related works in this area.

- [1] Zhu L, Li Y, Liu N, et al. M&m vto: Multi-garment virtual try-on and editing[C]//Proceedings of the IEEE/CVF Conference on Computer Vision and Pattern Recognition. 2024: 1346-1356.
- [2] He Z, Ning Y, Qin Y, et al. VTON 360: High-fidelity virtual try-on from any viewing direction[C]//Proceedings of the Computer Vision and Pattern Recognition Conference. 2025: 26388-26398.
- [3] Li Y, Zhou H, Shang W, et al. Anyfit: Controllable virtual try-on for any combination of attire across any scenario[J]. Advances in Neural Information Processing Systems, 2024, 37: 83164-83196.
- [4] Velioglu R, Bevandic P, Chan R, et al. MGT: Extending Virtual Try-Off to Multi-Garment Scenarios[C]//Proceedings of the IEEE/CVF International Conference on Computer Vision. 2025: 6039-6048.
- [5] He Z, Chen P, Wang G, et al. Wildvidfit: Video virtual try-on in the wild via image-based controlled diffusion models[C]//European Conference on Computer Vision. Cham: Springer Nature Switzerland, 2024: 123-139.

---

> ### Author Response · Authors · 2025-11-24
> **Response (1/2)**
>
> **Q1:** Limited Input Scalability.
>
> **A1:** Thank you for your insightful comment on the method's scalability. **Our primary goal is to address the challenging task of customizing a single human character from multiple reference subjects, for which the four input categories (face, upper garment, lower garment, and bag) are the most common and essential.** At inference time, users are free to select any combination of the four reference types, as depicted in Figure 1.
>
> **The significance of the binding between clothing categries and positional embeddings is to faciliates the learning of more specific, category-related attributes, which extends beyond merely general reference capability.** This is particularly crucial for clothing, which exhibit complex non-rigid deformations and wearing styles. The lack of fidelity in clothing-related details in the results of prior work [1][2] serves as an indirect indication of this point.
> Therefore, while we do not explicitly train for multiple same-category subjects, our framework exhibits **a certain generalization for categories that share similar wearable interactions with the human body**. For instance, to generate an image with two upper garments, we can provide a second 'upper garment' reference using the positional embedding originally designated for the 'lower garment'. The generation results are shown in **Figure 22 in Scetion G.5.1 of the Appendix**.
> As shown by the results, while the overall structure is synthesized correctly, there are subtle losses of consistency in specific details, such as the collar and hem, **further validating that our category-position binding is essential for robust consistency.**
>
> Regarding scalability to new object types (e.g., "additional accessories"), **our framework is designed for straightforward extension**. One can either introduce a new condition ID for a new category or group a new accessory with an existing category that shares a similar wearable interactions. Then, this extension can be achieved by fine-tuning on readily available single-subject data, circumventing the need for complex, manually curated multi-subject datasets. **Figures 24 and 25 in Section G.5.2 of the Appendix** demonstrates some generation cases on additional accessories, such as hat and shoes.
>
> In summary, while our current implementation is deliberately scoped, the underlying framework demonstrates both inherent flexibility and a clear, data-efficient path toward future scalability.
>
> [1] Mou C, Wu Y, Wu W, et al. Dreamo: A unified framework for image customization[J]. arXiv preprint arXiv:2504.16915, 2025.
>
> [2] Wu C, Zheng P, Yan R, et al. OmniGen2: Exploration to Advanced Multimodal Generation[J]. arXiv preprint arXiv:2506.18871, 2025.
>
> **Q2:** Please provide more details on the internal dataset used in the paper. Clarifying its specific characteristics would benefit the community as a whole.
>
> **A2:** Thank you for raising this important point regarding data transparency and reproducibility. We completely agree that this is essential for a fair evaluation.
> The absence of an available open-source dataset for our specific task of human-centric multi-subject customization necessitates the collection of a new evaluation benchmark. As described in **Section 4.1 and Section C in the Appendix**, we construct the HumanBench by supplementing two public datasets (DressCode and DeepFashion) with a small set of internal data. The introduction of internal data is necessary to overcome the limitations of the two public datasets, e.g., the lack of diversity in head poses and races, simple garment textures, and scarcity of bag images. **The internal data specifically introduces these more challenging and diverse examples across human races, garment styles and textures, and bag types. The final benchmark comprises 100 face-clothing pairs.**
> To guarantee reproducibility and to facilitate future research in this area, we will make the HumanBench publicly available upon acceptance of this paper, as stated in the **Reproducibility Statement** following the main manuscript. We believe that releasing this benchmark will be a valuable contribution to the community.
>
> **Q3:** Is the proposed DAM module trained only on the single-subject dataset? From Fig. 5, it seems that the input is limited to Q.
>
> **A3:** Yes, the DAM module is trained end-to-end only on the single-subject dataset. For each query token of Q, it functions as a modulator that predicts a single re-weighting score. This mapping is learned by seeking to minimize the image reconstruction objective. **In essence, the model learns fine-grained local correspondences by reweighting multiple reference tokens against each target query token. This mechanism is thus agnostic to the total number of reference subjects**.

---

> ### Author Response · Authors · 2025-11-24
> **Response (2/2)**
>
> **Q4:** How does the module infer the required weights solely from the target image? Furthermore, how does it generalize to a larger number of reference items? Are the re-weighting scores computed independently for each item?
>
> **A4:** Thank you for this thoughtful comment. **The DAM module learns a general policy: for any target query token in regions with competing features, predict a low re-weighting score to force a decisive choice. And no, the re-weighting scores are not computed independently for each reference item.** The key is that the module learns to resolve a more general problem: feature ambiguity, which is abundant even in the single-subject training data. This ambiguity arises from feature conflicts between the subject and its background, or between fine-grained details within the subject itself (e.g., hair vs. face). In these cases, any feature blending leads to high reconstruction loss.
> During multi-subject inference, the conflict between different subjects simply presents as a new, more intense form of this familiar ambiguity. **The pre-trained DAM applies the same general policy: it outputs a low temperature score to reweight the relevance of multiple reference tokens, which forces the softmax function to make a highly decisive selection.** This amplifies minor attention differences, allowing the model to decisively resolve feature conflicts, which we describe as "implicit feature routing for each query token" in Section 3.4. This is how the strategy, learned on simpler forms of conflict, naturally generalizes to solve the more complex multi-subject case.
>
> **Q5:** To strengthen the argument, it would be more convincing to include objective results that validate the effect of the position-aware conditioning strategy. I may have overlooked it, but I only found visual comparisons in Figure 9.
>
> **A5:** Thank you for this valuable suggestion. We agree that the inclusion of this quantitative ablation study makes the validation of our proposed strategy more complete and rigorous.
> The table below reports the quantitative comparisons, where we compare with two alternative implementations: **Var. 1** that is based on optimizable index embeddings and **Var. 2** that utilizes offsets in the virtual temporal dimension. As can be seen from the results, the position-aware conditioning strategy (**Var. 3**) yields **an obvious improvement across consistency-related metrics, highlighted by a 0.3-point increase in the face similarity score**. This can be attributed to the **strong spatial inductive bias introduced by our category-position binding**, which offers a more concrete and distinguishable signal than optimization-reliant index embeddings and the subtle offset in the virtual temporal dimension.
> These quantitative findings are fully consistent with our qualitative results.
>
> Quantitative ablations of the position-aware conditioning strategy. Bold indicates the optimal result.
>
> | **Method** | **FaceSim** | **GME-I** | **GME-T** | **FashionCLIP-I** | **FashionCLIP-T** | **DINO** |
> | :--- | :---: | :---: | :---: | :---: | :---: | :---: |
> | Var.1 | 0.298 | 0.602 | 0.684 | 0.623 | 0.341 | 0.367 |
> | Var.2 | 0.297 | 0.595 | 0.681 | 0.613 | 0.340 | 0.340 |
> | Var.3 | **0.599** | **0.634** | **0.695** | **0.625** | **0.344** | **0.392** |
>
> **Q6:** The visual comparison figures (e.g., Fig. 6, Fig. 9, Fig. 10) do not display the corresponding input text of the generated images.
>
> **A6:** Thank you for this kind reminder. We have now added the corresponding input text of the generated images in Fig. 6, Fig. 9, and Fig. 10. Please refer to the revised manuscript for details.
>
>
> **Q7:** In several visual comparisons (e.g., Fig. 9, Fig. 10), the differences between methods are not very clear. I suggest highlighting the key differences directly in the figures for better readability.
>
> **A7:** Thank you for this helpful suggestion. We agree that highlighting the key differences would improve clarity. Following the advice, we have now added colored arrows to the images in Fig. 6, Fig. 9, and Fig. 10 to highlight the key differences. Please refer to the revised manuscript for details.
>
> **Q8:** The proposed human-centric multi-subject customization task is very closely related to the multi-garment virtual try-on task. Therefore, it would be beneficial to discuss more related works in this area.
>
> **A8:** Thank you for the valuable suggestion. We have added a discussion of the recommended related works in **Section 2.2** of our revised manuscript.

---

> ### Comment · Reviewer_7JcY · 2025-11-26
> **Response to Author**
>
> Thank you for the detailed response. It addresses several of my earlier concerns. In particular, the clarifications regarding the DAM training setup (Q3), the computation of re-weighting scores (Q4), the new objective evaluations supporting the position-aware conditioning strategy (Q5), and the expanded discussion of related work (Q8) are informative and satisfactory.
>
> However, I still have several remaining concerns:
>
> - ***Input categories and scalability***.
> You argue that the four input categories (face, upper garment, lower garment, and bag) are the “most common and essential.” I remain unconvinced that these categories sufficiently cover the scope of human-centric multi-subject customization, as accessories such as watches or glasses are also frequent and practically important. However, as this is the first work to explicitly tackle this problem and given that the framework can be trained solely on single-subject data, I believe using these four input categories is somewhat acceptable for validating the proposed approach.
>
> - ***Training data transparency and potential distribution bias***.
> I initially misunderstood the usage of internal data, but upon re-reading the paper, I realized that the description of ***training data sources is missing or at least unclear***. It is important to explicitly specify which datasets are used for training and which (including HumanBench and its components) are used solely for evaluation.
> ***This leads to a further concern***: the strong performance on the test set may partially result from a closer match between your training and evaluation distributions, particularly for categories such as bags that competing methods do not model. Please clarify (1) what training datasets are used and (2) whether your advantages remain when “rare” or less common categories (e.g., bags) are excluded to enable a more comparable evaluation across methods. I encourage you to provide a clear description of the training data and, if possible, a fairer comparison limited to shared categories.
>
> - ***Figures, prompts, and captions***.
> Although some figures have been updated, several visual examples still omit the corresponding text prompts. Since text–image alignment is a key aspect of the task, prompts should be shown alongside the generated images. Additionally, please ***proofread all figure and table captions ***; for instance, the caption for Fig. 26 appears incorrect. All these issues should be carefully fixed before publication.

---

> > ### Author Response · Authors · 2025-11-27
> > **Response**
> >
> > **Q1:** Input categories and scalability.
> >
> > **A1:**
> > Thank you for this insightful and encouraging comment. We fully agree with your assessment and are grateful for your understanding.
> > As you kindly point out, **our primary goal in this work is to propose and validate a new, data-efficient attempt at this challenging human-centric multi-subject customization problem. Our core contribution is a framework that makes this very extension process practical and feasible, by avoiding the need for complex multi-subject datasets.** Experiments on the four existing categories, along with supplementary experiments on new categories like hats and shoes (**Figs. 24 and 25**), both demonstrate that our method is not limited to a specific set of input categories. By using only easily available single-subject data, our model can be efficiently extended to new human-centric subjects.
> >
> > Therefore, we sincerely appreciate your perspective that our approach—which combines a robust generalization baseline with a practical, data-efficient extension mechanism—is a valuable and acceptable validation of a new direction. We believe this work lays a crucial and scalable foundation for more comprehensive systems in the future.
> >
> > **Q2:** Training data transparency and potential distribution bias.
> >
> > **A2:**
> > Thank you for pointing out this critical issue. Our training set consists of real-world, single-subject data pairs across four reference types: face, upper garment, lower garment, and bag, as illustrated by the green box examples in the middle of Figure 2. Specifically, the dataset comprises 40k pairs for face, 50k for upper garments, 40k for lower garments, and 100k for bags (with the larger volume for bags intended to account for their diverse carrying styles). To ensure data balance during training, we employ a resampling ratio of 2:2:2:1 for the face, upper-body, lower-body, and bag categories, respectively. **Crucially, to maintain experimental fairness, we strictly sample the training and evaluation data from distinct sources, thereby ensuring there is no distribution bias.** To further clarify the composition of HumanBench, the table below details the data source proportions of each data category used for model evaluation (also shown in Table 5 in the latest revised manuscript).
> >
> > | **Source** | **Face** | **Upper** | **Lower** | **Dress** | **Bag** |
> > | :--- | :---: | :---: | :---: | :---: | :---: |
> > | DeepFashion | 25 | 0 | 0 | 0 | 0 |
> > | DressCode | 0 | 43 | 34 | 10 | 0 |
> > | Internal | 25 | 7 | 6 | 0 | 20 |
> > | Total | 50 | 50 | 40 | 10 | 20 |
> >
> >
> > Furthermore, to further validate the effectiveness of our method, we conduct a "fairer comparison limited to shared categories" as suggested,  i.e., without bag inference. The evaluation set for this experiment consists solely of face images from the public DeepFashion dataset and clothing images from the DressCode dataset, comprising 50 image pairs across 3 different scenes for a total of 150 pairs. Consistent with the main experiment, we report the results averaged over 4 seeds.
> > **Table 6 and Figure 27 in Appendix G.6** of the latest manuscript present the quantitative and qualitative results, respectively. As can be seen, **for in-distribution categories, previous methods still exhibit poor adherence to subject consistency**. Moreover, these findings suggest that the performance degradation in these previous methods stems from their failure to address the feature competition that arises from multiple reference inputs, rather than to limitations from not having seen a particular data category. Therefore, **the strength of our method lies in its ability to generate consistent details in multi-subject reference scenarios, and not just in the introduction of the new bag category**.
> > | **Method** | **FaceSim** | **GME-I** | **GME-T** | **FashionCLIP-I** | **FashionCLIP-T** | **DINO** |
> > | :--- | :---: | :---: | :---: | :---: | :---: | :---: |
> > | UNO | 0.528 | 0.718 | 0.699 | 0.700 | 0.304 | 0.501 |
> > | DreamO | 0.694 | 0.693 | **0.705** | 0.658 | 0.306 | 0.454 |
> > | OmniGen2 | 0.538 | 0.709 | 0.700 | 0.704 | 0.303 | 0.483 |
> > | Ours | **0.732** | **0.736** | 0.696 | **0.739** | **0.312** | **0.523** |
> >
> > This result further highlights the scalability advantages of our proposed single-subject training paradigm. Thank you again for your valuable suggestions, which are crucial for improving the paper's integrity.
> >
> >
> > **Q3:** Figures, prompts, and captions.
> >
> > **A3:**
> > Thank you for your careful review. We sincerely apologize for this oversight. We have now supplemented visual results with their corresponding text prompts. Furthermore, we have conducted a thorough review of all figure and table captions throughout the manuscript and have corrected the errors from the previous version. Thank you again for your attention to detail and thoroughness. Your feedback has significantly improved the clarity and quality of our paper.

---

> > > ### Comment · Reviewer_7JcY · 2025-11-28
> > > **Response**
> > >
> > > Thank you for your timely response and the additional experiments. I appreciate the clarifications and believe these revisions will further improve the paper’s quality. However, information regarding the training data (its ***sources***, ***construction process***, and ***accessibility to the community***) remains unclear. Since the proposed framework’s contribution is closely tied to the availability and transparency of the underlying data and code, I strongly encourage the authors to provide a clearer, more explicit description of the datasets used.
> > >
> > > Finally, I still believe that relying solely on single-subject datasets is practically valuable, especially given the difficulty of collecting high-quality multi-subject datasets. This direction could be a meaningful complement to data-scarce scenarios, and I look forward to seeing more applications of this strategy in future work. I sincerely hope that the AC and other reviewers will consider the practical advantages of this approach and reconsider the overall score.

---

> > > > ### Author Response · Authors · 2025-11-28
> > > > **Response**
> > > >
> > > > We greatly appreciate your supportive comments, which serve as a significant source of encouragement for us. As requested, we provide **a more explicit description of our training data** here. The training data is sourced from e-commerce websites, featuring a rich diversity of subjects in various human-centric scenarios. The construction process begins by first gathering all available images for a given subject, followed by filtering them based on category label to retain only those that explicitly contain the subject. These filtered images are then formed into pairs, and a Vision-Language Model (VLM) is employed to verify that the image pairs depict the same subject, ensuring identity consistency. By repeating this process for each reference type, we collect image sets of the same subject from various viewpoints and scenarios. These collected images then undergo a series of preprocessing steps, i.e., cropping and segmentation, to obtain the final reference and target image pairs. Finally, the reference and target images are captioned respectively to produce the "item prompt" and "full prompt" required for model training.
> > > >
> > > > To ensure accessibility and transparency for the community, we will make the datasets, along with our model, publicly available upon the acceptance of this paper, and we will update details regarding the training data in the Appendix. Thank you again for acknowledging the practicality and data-efficient nature of our work.

---

### Official Review · Reviewer_ZR1h · 2025-10-31

**Soundness:** 3
**Presentation:** 3
**Contribution:** 2
**Rating:** 4
**Confidence:** 4

**Summary:**

This paper presents One-for-All, which is a framework for human image generation that combines multiple subjects like face, clothes, and bags using only single-subject training data. It introduces a position-aware learning strategy that gives each subject a fixed spatial location to keep them consistent. Two key designs help the model work well: a cross-modal position module to align text and image meaning, and a dynamic attention module to balance focus among subjects. The experiment results show that One-for-All produces more consistent and realistic results than the existing methods.

**Strengths:**

1. The proposed position-aware learning and attention design help keep all subjects well aligned and consistent in appearance.
2. The paper builds a new benchmark and shows strong improvements over several state-of-the-art methods in both quality and consistency.
3. The visualization of the attention after adding DAM demonstrates the reason for improvement which is intuitive.

**Weaknesses:**

1. The paper relies only on automatic metrics to measure image quality and consistency. Without human evaluation, it is unclear whether the results align with real user perception.
2. he study of the position-aware conditioning strategy shows only qualitative results, with limited images and no quantitative scores, which makes the contribution less convincing.
3. The experiments focus mainly on faces and clothes, without testing on other human-related or non-human subjects, so the generalization of the method remains uncertain.

**Questions:**

Since the paper focuses on human-centric generation, perceptual quality is important. Have the authors considered conducting a user study or pairwise preference test to verify whether their results are visually preferred by human observers?

---

> ### Author Response · Authors · 2025-11-24
> **Response (1/2)**
>
> **Q1:** User study to verify whether the results align with real user perception.
>
> **A1:** Thank you for this important comment. We completely agree that human evaluation is crucial for validating perceptual quality and consistency. To this end, we have conducted a user study to complement our automatic metrics. **The assessment is performed across the following key dimensions: subject consistency, text fidelity concerning the subject, text fidelity concerning the background, and overall visual fidelity**. For detailed methodology and results, please refer to **Table 3 in Section 4.4**. The results confirm that users largely prefer our method for its superior subject consistency compared to the baselines. These findings are in strong alignment with the results from our automatic metrics.
>
> User study of the baseline methods. Bold indicates the optimal result.
> | **Method** | Subject Consistency | Text Fidelity (subject) | Text Fidelity (background) | Visual Fidelity |
> | :--- | :---: | :---: | :---: | :---: |
> | Kontext | 0.17 | 0.23 | 0.18 | 0.22 |
> | MS-Diffusion | 0.04 | 0.04 | 0.03 | 0.02 |
> | UNO | 0.09 | 0.12 | 0.20 | 0.19  |
> | DreamO | 0.10 | 0.09 | **0.22** | 0.17 |
> | OmniGen2 | 0.19 | 0.18 | 0.17 | 0.10 |
> | Ours | **0.41** | **0.34** | 0.20 | **0.30** |
>
>
> **Q2:** The study of the position-aware conditioning strategy shows only qualitative results, with limited images and no quantitative scores, which makes the contribution less convincing.
>
> **A2:** Thank you for this valuable suggestion. We agree that the inclusion of this quantitative ablation study contributes to a more complete validation of our proposed strategy.
> The table below reports the quantitative comparisons, where we compare with two alternative implementations: **Var. 1** that is based on optimizable index embeddings and **Var. 2** that utilizes offsets in the virtual temporal dimension. As can be seen from the results, the position-aware conditioning strategy (**Var. 3**) yields **an obvious improvement across consistency-related metrics, highlighted by a 0.3-point increase in the face similarity score**. This can be attributed to the **strong spatial inductive bias introduced by our category-position binding**, which offers a more concrete and distinguishable signal than optimization-reliant index embeddings and the subtle offset in the virtual temporal dimension.
> These quantitative findings are fully consistent with our qualitative results.
>
> Quantitative ablations of the position-aware conditioning strategy. Bold indicates the optimal result.
>
> | **Method** | **FaceSim** | **GME-I** | **GME-T** | **FashionCLIP-I** | **FashionCLIP-T** | **DINO** |
> | :--- | :---: | :---: | :---: | :---: | :---: | :---: |
> | Var.1 | 0.298 | 0.602 | 0.684 | 0.623 | 0.341 | 0.367 |
> | Var.2 | 0.297 | 0.595 | 0.681 | 0.613 | 0.340 | 0.340 |
> | Var.3 | **0.599** | **0.634** | **0.695** | **0.625** | **0.344** | **0.392** |

---

> ### Author Response · Authors · 2025-11-24
> **Response (2/2)**
>
> **Q3:** The experiments focus mainly on faces and clothes, without testing on other human-related or non-human subjects, so the generalization of the method remains uncertain.
>
> **A3:** Thank you for your insightful comment on the model's generalization. To evaluate the model’s generalization to other human-centric reference categories, we present its generation results with a face and a scarf as references, a category unseen during training (**Figure 21, Appendix G.5.1**).
>
> **The results reveal that the model possesses a foundational generalization capability.** In this scenario, where category-specific priors are naturally not applied, the model successfully composes the unseen scarf onto the person. It demonstrates that the model learns the broad concept of "wearable styles" of categories rather than overfitting to specific clothes like 'shirts'.  To further demonstrate the generalization of our method with more human-related accessories, we add supplementary experimental results. These new results, featuring additional 'hat' and 'shoes' as reference subjects, can be found in **Figures 24 and 25 in Section G.5.2 of the Appendix**.
>
> Moreover, inspired by the comment, we conduct an extension experiment to test model generalization to non-human subjects. Specifically, we introduce a generic 'other' category (assigned to a condition ID: 5) and train the model on general single-subject data from the public Subject200k [1] dataset. Model performance is evaluated on the DreamBench [2]. The generation results are presented in **Figure 26 in Section G.5.3 of the Appendix**. As shown, our model readily acquires the ability to generate these diverse non-human subjects with high consistency. This result strongly supports that our framework is domain-agnostic and robustly applicable beyond the primary human-centric focus.
>
> [1] Tan Z, Liu S, Yang X, et al. Ominicontrol: Minimal and universal control for diffusion transformer[C]//Proceedings of the IEEE/CVF International Conference on Computer Vision. 2025: 14940-14950.
>
> [2] Ruiz N, Li Y, Jampani V, et al. Dreambooth: Fine tuning text-to-image diffusion models for subject-driven generation[C]//Proceedings of the IEEE/CVF conference on computer vision and pattern recognition. 2023: 22500-22510.

---

### Author Response · Authors · 2025-11-24
**Global Response**

We sincerely thank all reviewers for their insightful feedback and constructive suggestions. We are encouraged by the positive feedback, particularly the recognition of our **novel and effective method** for learning multi-subject consistency from single-subject data (7JcY, fHmp, iKbq), the **intuitive design and validation** of our proposed modules (ZR1h, fHmp), the **strong experimental results** that validate our superiority on HumanBench (ZR1h, fHmp), and the overall high quality of the work, including its **clear presentation and interesting task definition** (7JcY, iKbq).

Through comprehensive revisions and extensive new experiments, we are confident that we have fully addressed all concerns and strengthened our paper's core contribution: **a novel and data-efficient framework for learning multi-subject consistency from single-subject data, offering a practical and robust solution for human-centric multi-subject customization**. Key updates include:

*   **Human Evaluation** (ZR1h): We have conducted a user study (Tab. 3) that confirms a strong user preference for the subject consistency of our method, which aligns well with the automatic metrics in Tab. 1.
*   **New Quantitative Ablations** (ZR1h, 7JcY): We now add a quantitative ablation study to further validate the effectiveness of our position-aware conditioning strategy in Tab. 4.
*   **New Generalization & Scalability Experiments** (ZR1h, 7JcY, fHmp): To demonstrate the generalization and scalability of our framework, we add new generation results in Appendix G:
    *   Generalization to unseen cases like 'scarf', 'multi-top', and 'cartoon character' (Figs. 21, 22, 23).
    *   Scalability with up to 5 and 6 subjects at once (Figs. 24, 25).
    *   Extension to non-human subjects on the DreamBench dataset (Fig. 26).
*   **Comparisons with Closed-Source Models** (fHmp): We now include qualitative comparisons with closed-source models like GPT-4o, Qwen-Image-Edit and Nano Banana, highlighting our method's advantage in human-centric multi-subject consistency (Fig. 20).
*   **Fairer Comparisons on Shared Categories** (7JcY): We now include a "fairer comparison" experiment limited to shared categories (face and clothing), excluding bags, to isolate our method's core strengths. The new quantitative (Tab. 6) and qualitative (Fig. 27) results demonstrate that our method's advantage lies in its robustness in maintaining multi-subject consistency, not just in the introduction of a new category.
*   **Detailed Data Composition and Transparency** (7JcY): We provide a detailed description of our training data (Appendix H), including data sources, construction process, and composition. We further clarify the composition of our evaluation set, HumanBench, in a new table (Tab. 5) and affirm that training and evaluation sets are sourced from distinct distributions to ensure fairness.
*   **Manuscript and Figure Enhancements** (7JcY, fHmp): We update figures with text prompts and colored arrows for improved clarity (e.g., Figs. 6, 9, 10) and expand our discussion of related works (Sec. 2.2).
*   **Clarification of Core Modules (CCPA & DAM)** (7JcY, iKbq): We clarify how our core modules work: CCPA establishes targeted cross-modal synergy through text-image centroid position alignment (Fig. 10, Tab. 1), while DAM prevents feature blending by triggering a "winner-takes-all" attention state in response to high-conflict queries (Figs. 10, 11, Tab. 1).

Point-by-point responses to each reviewer are provided below. All revisions in the latest manuscript are marked in **blue**.

Finally, we would like to express our sincere gratitude to Reviewer 7JcY for the timely and insightful follow-up during the disscusion period. We are greatly encouraged by **the recognition of our work's practical value and data efficiency, and the recommendation to "reconsider the overall score"**. The characterization of our approach as a **"meaningful complement to data-scarce scenarios"** captures a core motivation for our work. This validation underscores the importance of addressing the well-known challenges in multi-subject data collection, and we hope our work opens a practical pathway for developing new solutions to data-scarce multi-subject generation tasks.

---

### Meta-Review · Area_Chair_PAB2 · 2026-01-05

**Summary:**

The reviewers agree that the paper addresses a practically important problem and presents a technically sound framework for learning multi-subject consistency from single-subject data. The motivation is clear, and the empirical results indicate consistent improvements over prior methods.

However, reviewers raised several concerns that limit confidence in a clear accept. In particular,
1. the novelty of the proposed modules was viewed as incremental relative to existing position-based control and attention modulation techniques.   The idea of using positional embeddings to handle multi-object reference conflicts is already well-established, making this component less distinctive in contribution.


2. Experimental results are limited and not generalized too diverse scenes/conditions or large-scale datasets

Overall, this is a well-executed but borderline submission, with strengths in practicality and experimental effort, balanced against lingering concerns about conceptual novelty and scope.

**Reviewer Concerns:**

Though authors are trying to resolve the reviewers concerns but the main concerns on technical novelty will limit the score of this paper. Since the proposed novelty is very limited and initial submission is missing many important experiments, I plan to reject this paper

**Reviewer Scores:**

This paper got three reject (4) and 1 accept (6)

---

### Decision · Program_Chairs · 2026-01-26

Reject